# Location-agnostic site-specific protein bioconjugation via Baylis Hillman adducts

Mudassir H. Mir[1,2], Sangeeta Parmar[1,2], Chhaya Singh [1] & Dimpy Kalia [1] ✉

Proteins labelled site-specifically with small molecules are valuable assets for chemical biology and drug development. The unique reactivity profile of the 1,2-aminothiol moiety of *N*-terminal cysteines (*N*-Cys) of proteins renders it highly attractive for regioselective protein labelling. Herein, we report an ultrafast *Z*-selective reaction between isatin-derived Baylis Hillman adducts and 1,2-aminothiols to form a bis-heterocyclic scaffold, and employ it for stable protein bioconjugation under both in vitro and live-cell conditions. We refer to our protein bioconjugation technology as Baylis Hillman orchestrated protein aminothiol labelling (BHoPAL). Furthermore, we report a lipoic acid ligase-based technology for introducing the 1,2-aminothiol moiety at any desired site within proteins, rendering BHoPAL location-agnostic (not limited to *N*-Cys). By using this approach in tandem with BHoPAL, we generate dually labelled protein bioconjugates appended with different labels at two distinct specific sites on a single protein molecule. Taken together, the protein bioconjugation toolkit that we disclose herein will contribute towards the generation of both mono and multi-labelled protein-small molecule bioconjugates for applications as diverse as biophysical assays, cellular imaging, and the production of therapeutic protein–drug conjugates. In addition to protein bioconjugation, the bis-heterocyclic scaffold we report herein will find applications in synthetic and medicinal chemistry.

Proteins appended site-specifically with small molecules enable powerful applications in both basic biological sciences and drug development[1–4]. Prominent examples include the use of proteins labelled with fluorescent dyes for cellular imaging[5–7], and antibodies appended with drugs as promising anti-cancer agents[8–10].

Developing efficacious protein bioconjugation reactions is challenging because of the stringent pre-determined conditions under which they need to be performed[11]. Indeed, in contrast to typical organic reactions that can be subjected to extensive optimisation by altering parameters such as solvent and temperature to minimise side-product formation and enhance product yields and reaction rates, bioconjugation needs to be performed under physiological conditions (at room temperature in aqueous solutions at near neutral pH) that cannot be altered without compromising cellular integrity and/or

protein stability. Another indispensable requirement for an ideal protein bioconjugation reaction is that it should proceed site-specifically on proteins without disrupting protein function to yield homogenous conjugates[4,12,13].

Despite the development of several organic reactions for site-specific protein bioconjugation[1,4,9,12], most existing bioconjugation platforms entail the use of elevated concentrations of labelling reagents to compensate for their slow reaction kinetics under physiological conditions. These reaction conditions are not ideal as they encourage protein aggregation and precipitation[11,14], and can lead to reagent-induced cytotoxicity[2,5,6]. Consequently, developing reactions that proceed rapidly under physiological conditions with stoichiometric low-micromolar protein and labelling reagent concentrations to quantitatively yield stable protein bioconjugates is highly desirable.

[1]Department of Chemistry, Indian Institute of Science Education and Research (IISER) Bhopal, Bhopal Bypass Road, Bhauri, Bhopal 462066 Madhya Pradesh, India. [2]These authors contributed equally: Mudassir H. Mir, Sangeeta Parmar. ✉e-mail: dimpy@iiserb.ac.in

Cysteine-mediated thiol modification has emerged as one of the most promising approaches for protein bioconjugation[15–19]. N-terminal cysteines (N-Cys) are particularly suited for site-specific protein labelling due to the unique and narrow reactivity profile of their 1,2-aminothiol moiety, coupled with their low frequency of occurrence in the proteome[15,20,21]. Although several promising N-Cys-mediated protein bioconjugation approaches have been developed, they possess limitations. For example, N-Cys labelling via 2-cyanobenzothiazole (CBT) demonstrates moderate reaction kinetics and shows cross-reactivity with internal Cys residues of proteins[22]. The use of benzaldehyde-based reagents for labelling N-Cys to form cyclic thiazolidine adducts is also not ideal as it entails two day-long treatments of proteins with a large excess of reagents under acidic conditions[23,24]. Appending an o-boronic acid substituent to benzaldehyde substantially enhances the labelling rate but the resulting chelated thiazolidino-boronate (TzB) conjugates lack stability[25,26]. Although introducing an ester moiety adjacent to the formyl group in these substrates leads to the formation of stable N-acyl thiazolidine boronic acid linkages, its formation proceeds via a TzB intermediate that undergoes a slow (several hours-long) acyl transfer step to form the final product[27]. Slow reaction kinetics also limits the use of the classical native chemical ligation chemistry that involves the reaction of N-Cys with thioesters[28,29], and malononitrile (TAMM)-based 1,2-aminothiol derivatizing reagents[30]. A recently reported cyclopropenone-mediated Michael addition-cyclization reaction to modify N-Cys is also limited due to its moderate rates, and because it yields a heterogeneous mixture of diastereomeric cyclic adducts[31].

Apart from the drawbacks of the individual approaches described above, an intrinsic general limitation of 1,2-aminothiol-mediated protein bioconjugation via N-Cys is that it is not applicable for labelling sites other than the N-termini of proteins. This limitation has been addressed by introducing 1,2-aminothiol containing non-natural amino acids into the desired bioconjugation sites[30,32–34]. The amber codon suppression technology that was employed in these reports for non-natural amino acid incorporation, however, is technically convoluted[35,36] and a more straightforward approach for installing 1,2-aminothiols into proteins would be desirable.

Herein, we report an unprecedented isatin-derived Baylis Hillman (IBH) adduct-mediated 1,2-aminothiol derivatization reaction that proceeds rapidly (rate constant>$10^3$ M$^{-1}$s$^{-1}$) under physiological conditions to yield a stable C=C linked bis-heterocyclic scaffold containing the thiomorpholine and indolinone cores. We employ this chemistry to develop a protein bioconjugation platform that we refer to as BHoPAL (Baylis Hillman orchestrated protein aminothiol labelling) that enables quantitative protein labelling with low/sub-micromolar protein and labelling reagent concentrations (Fig. 1a). Furthermore, we report a lipoic acid ligase-based technology for installing the 1,2-aminothiol moiety at any desired site on proteins, rendering BHoPAL and other 1,2-aminothiol bioconjugation platforms location-agnostic, and not limited to labelling only the N-termini of proteins (Fig. 1b). In addition to bioconjugation, the bis-heterocyclic scaffold we report herein will open doors to applications in medicinal and synthetic organic chemistry.

## Results

### Development of the 1,2-aminothiol–IBH adduct conjugation chemistry

Baylis Hillman adducts are versatile synthons in organic synthesis because they present several suitable chemical handles for derivatization[37–41]. Isatin-derived Baylis Hillman (IBH) adducts have been utilised as scaffolds for the synthesis of complex natural products[42,43], drug intermediates[44–47], and clinical pharmaceuticals[44,48]. Despite this extensive body of work on IBH adducts, to the best of our knowledge, their reactivity with 1,2-aminothiols has not been reported.

We envisioned that IBH adducts containing two appropriately located leaving groups, an α-acetate group and a β-cyano group (1, Fig. 2a), would undergo two sequential addition-elimination reactions with 1,2-aminothiols to generate 6-membered cyclic adducts

**a** N-Cys protein bioconjugation via BHoPAL

➤ Ultrafast kinetics (k>$10^3$ M$^{-1}$s$^{-1}$) under physiological conditions

➤ Quantitaive yields with low/sub-μM protein and reagent concentrations

➤ Stable bioconjugates

**b** Incorporation of the 1,2-aminothiol moiety at any site in proteins

**Fig. 1 | The BHoPAL (Baylis Hillman orchestrated protein aminothiol labelling) strategy for the chemoselective bioconjugation of 1,2-aminothiols in proteins. a** Our approach employing isatin-derived Baylis Hillman (IBH) adduct-mediated 1,2-aminothiol derivatization of N-terminal Cys-containing proteins. The thiomorpholine ring of the conjugate is depicted in yellow and the indolinone ring in magenta. **b** Our strategy for installing the 1,2-aminothiol moiety at any site within proteins by employing the lipoic acid ligase (LplA) enzyme and thiazolidine-appended lipoic acid (TzLA) enzyme substrates. "LAP" is the LplA recognition peptide sequence that can be inserted at any desired bioconjugation site on the protein of interest. The thiazolidine ring is depicted in orange.

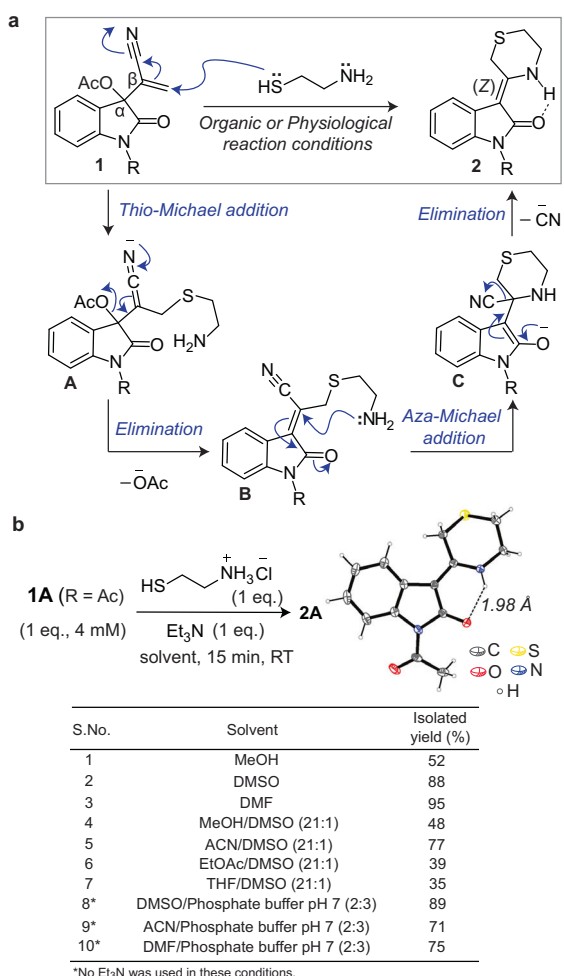

**Fig. 2 | Synthesis of C=C linked bis-heterocycles via IBH adduct-mediated 1,2-aminothiol derivatization. a** Proposed reaction mechanism. **b** Reaction of IBH adduct **1A** with cysteamine·HCl under various conditions. The ORTEP diagram for the single crystal structure of bis-heterocycle **2A** is shown on the right (CCDC no.: 2259911). The black dotted line denotes the N–H⋯O=C hydrogen bond.

(**2**, Fig. 2a). We hypothesised that this reaction would involve an initial thiol addition followed by the elimination of the acetate to generate intermediate **B**, which would then undergo an intramolecular aza-Michael addition followed by elimination of the cyano to form **2** (Fig. 2a). We further speculated that the alkene in **2** would be locked in the Z conformation due to an intramolecular N–H⋯O=C hydrogen bond (Fig. 2a), rendering this reaction highly Z-selective.

To test this hypothesis, we synthesised the N-acetyl IBH adduct **1A** (Supplementary Fig. 1) and treated it with cysteamine·HCl in the presence of a base (Et₃N, 1.0 eq.) for 15 min at room temperature (RT) in both protic and aprotic organic solvents (entries 1–7, Fig. 2b and Supplementary Fig. 3), as well as in aqueous solutions (entries 8–10). The desired product (**2A**) was obtained in good to moderate yields under all conditions tested including in neutral aqueous buffers wherein isolated yields of 71–89% were obtained, suggesting that this chemistry is biocompatible. As envisioned, the reaction was Z-selective, solely yielding the Z-alkene product **2A** (single crystal X-ray structure provided in Fig. 2b). The distance between the hydrogen atom of NH and the oxygen atom of O=C in this structure is 1.98 Å, consistent with the presence of an intramolecular hydrogen bond between them. Another N-acyl IBH adduct, **1B**, and the N-alkyl IBH adduct, **1C** (Fig. 3a) also gave the expected bis-heterocyclic scaffold upon treatment with cysteamine (compounds **2B** and **2C**, respectively), with the crystal structure of **2C** depicting an

identical intramolecular N–H⋯O=C hydrogen bond (Supplementary Fig. 4) as observed in **2A**. Moreover, this reaction was not limited to cysteamine as demonstrated by the formation of a similar cyclic bis-heterocycle upon the treatment of **1A** with another 1,2-aminothiol-containing molecule, L-Cys (**S2A-Cys**, Supplementary Fig. 3). To the best of our knowledge, such C=C linked bis-heterocycles containing the thiomorpholine and indolinone cores have not been previously reported.

The treatment of our N-acyl IBH adducts **1A** and **1B** (50 μM) with equimolar amounts of cysteamine·HCl under physiological conditions (in aqueous buffer solutions at pH 8 and 7) yielded the expected product quantitatively within 2 and 5 min, respectively (Fig. 3a). The reaction proceeded relatively slowly at pH 6, taking 45 min to proceed to completion (Fig. 3a). Replacing the N-acyl substituent with the N-alkyl group slowed down reaction kinetics with the N-methyl adduct **1 C** requiring 45 min, 2.5 h, and 5 h for quantitative conversions at pH 8, 7 and 6, respectively (Fig. 3a, right panel). The enhanced reactivity of N-acyl IBH adducts as compared to the N-alkyl adducts towards cysteamine is probably because of the increased electrophilicity of the carbonyl groups of intermediate **B** (Fig. 2a) formed with the former due to the electron-withdrawing effect of their N-acyl groups, thereby facilitating the aza-Michael addition reaction to form **C**.

Next, we studied the selectivity of our chemistry towards the 1,2-aminothiol moiety over the 20 proteinogenic amino acids starting with N-acetyl cysteine (NAC), a mimic of internal Cys residues of proteins. Incubating equimolar amounts of cysteamine, NAC, and the N-acyl IBH adduct **1A** at pH 6, 6.5, and 7 yielded the desired 1,2-aminothiol–IBH adduct conjugate, **2A**, in >90% yields in all three cases (Supplementary Fig. 9a). Replacing NAC with the other 19 natural amino acids yielded 85–99% yields of **2A** (Fig. 3b), establishing the excellent chemoselectivity of our chemistry towards 1,2-aminothiol. Similar results were obtained with our N-methyl adduct **1C** (Supplementary Figs. 9b and 10).

To more rigorously evaluate the selectivity of IBH adducts for 1,2-aminothiols over other alkane thiols, we treated **1A** with cysteamine in the presence of 1–5 eq. of either NAC or 1-hexanethiol at pH 6.5 and 8 (Supplementary Fig. 11). Excellent 1,2-aminothiol selectivity was observed in these reactions (>90% at pH 8 and as high as 97% at pH 6.5). These results demonstrate that the selectivity of **1A** for 1,2-aminothiol at nearly physiological pH is primarily governed by the higher concentration of the nucleophilic thiolate anion of the sulfhydryl group in the case of cysteamine (pKa ~8.2) as compared to that for NAC (pKa ~9.5) and 1-hexanethiol (pKa ~12)[21]. This rationale is analogous to the one that explains the selective labelling of the N-terminal α-amino group (pKa ~6–8) over lysine ε-amino group (pKa ~10) of proteins under slightly acidic aqueous conditions[49].

UV spectroscopy-based kinetic analyses demonstrated that the reaction between cysteamine and **1A** proceeds rapidly in aqueous solutions at pH 6, 7, 7.5 and 8 at RT, yielding second-order rate constants of 100, 1700, 3100 and 7600 M⁻¹s⁻¹ respectively (Fig. 3c). Similar results were obtained with the N-benzoyl adduct **1B** (Supplementary Fig. 12). These rates are substantially higher than those reported for 1,2-aminothiol derivatization by CBT (>300-fold)[22], cyclopropenone (>100-fold)[31], and TAMM (>700-fold)[30], and are similar to those reported for TzB conjugate formation[25]. To perform a head-to-head comparison between these 1,2-aminothiol derivatization reactions, we treated the IBH **1A**, cyclopropenone (CPO) and cyanobenzothiazole (CBT) reagents with cysteamine at neutral pH and at RT (both individually and in one pot). These experiments (Supplementary Fig. 13) revealed that the reactivity of **1A** with cysteamine supersedes that of the CPO and CBT reagents. Remarkably, incubating equimolar amounts of cysteamine and these three labelling reagents in one pot resulted in the quantitative formation of **2A** (the cysteamine conjugate of **1A**) within 5 min, while yielding negligible amounts of the cysteamine conjugates of CPO and CBT (Supplementary Fig. 13b).

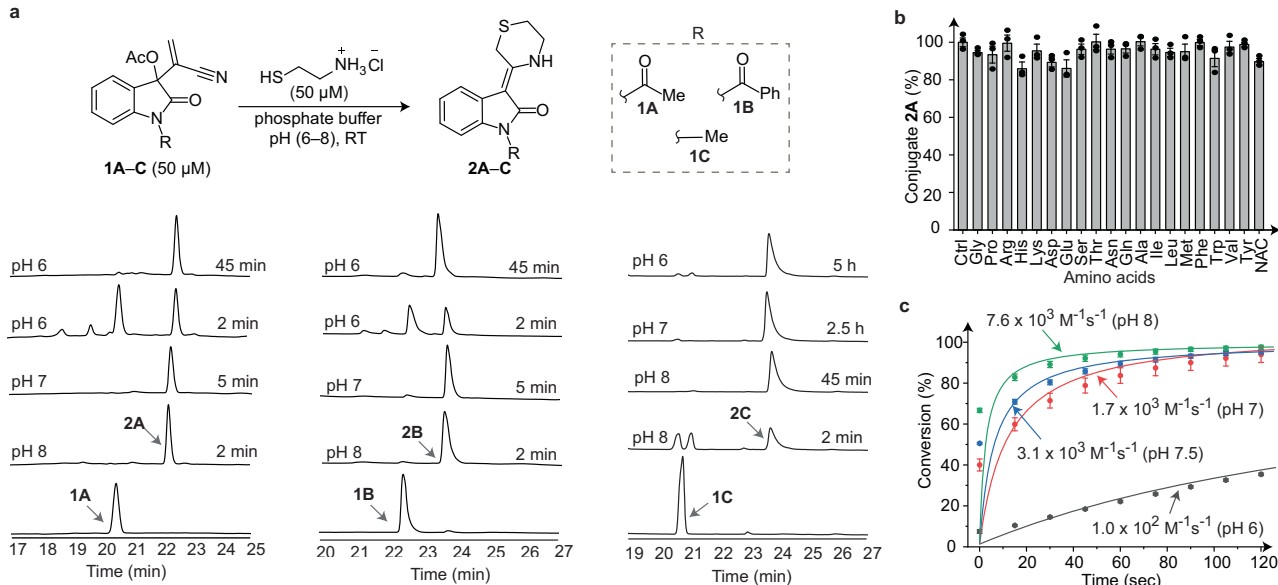

**Fig. 3 | Rapid and quantitative conjugation of *N*-acyl/alkyl IBH adducts with cysteamine under aqueous conditions. a** HPLC analysis of the reaction between **1A**–**C** and cysteamine at pH 6–8. **b** Chemoselective formation of conjugate **2A** in the presence of other amino acids. The reactions were performed by incubating equimolar concentrations (1 mM each) of **1A**, cysteamine, and other amino acids (individually) in a buffer at pH 6.5, at RT for 40 min followed by the quantitation of the product (**2A**) via HPLC (chromatograms depicted in Supplementary Fig. 10a). **c** Kinetics of formation of **2A**. The reactions were performed by incubating

equimolar concentrations (50 μM each) of **1A** and cysteamine in sodium phosphate buffer (pH 6–8) at RT, and monitored over time via UV–visible spectroscopy (traces depicted in Supplementary Fig. 12a). The second-order rate equation was fit to the data yielding the rate constants depicted in the plot. Each data point in **b** and **c** is an average of three measurements, and the error bars correspond to standard deviation values. Kinetics analyses on the formation of **2B** and **2C** are provided in Supplementary Fig. 12. All experiments were performed three times independently, and yielded similar results each time.

In addition to rapid kinetics, the stability of the conjugate under physiological conditions is of paramount importance for protein bioconjugation applications. In contrast to the TzB conjugates that lack stability in aqueous conditions[25], our IBH adduct conjugates **2A** and **2 C** retain integrity when incubated in aqueous solutions at pH 4–8 for 2–5 days at RT (Supplementary Fig. 14), and also upon exposure to excess thiols (10 mM *L*-cysteine over 1–2 days; Supplementary Fig. 15 and 5 mM glutathione over 3 days; Supplementary Fig. 16) at pH 7. The high stability of these conjugates together with the rapid rates of 1,2-aminothiol–IBH adduct conjugation under physiological conditions renders it highly attractive for protein bioconjugation.

## BHoPAL on pure proteins

With promising results on small molecules in hand, we proceeded to use our chemistry for labelling *N*-Cys residues of proteins. To achieve this goal, we produced the *N*-Cys variants of three model proteins: enhanced green fluorescent protein (eGFP), maltose binding protein (MBP) and mCherry, by recombinantly producing these proteins fused to the tobacco etch virus (TEV) protease substrate peptide (the ENLYFQC sequence) on their *N*-termini, and subjecting them to TEV protease cleavage (Fig. 4)[50]. Treating the resultant *N*-Cys proteins (10 μM) with 1.0 eq. of *N*-acyl IBH adducts **1A**, **1D** and **1E** in sodium phosphate buffer at pH 6.5 for 1 h at RT gave quantitative yields (>99%) for all the 9 protein conjugates as measured by ESI-MS analyses (Fig. 4, and Table 1 entries 1, 2 and 6 for eGFP; 22, 24 and 26 for MBP; 36, 38 and 40 for mCherry). Quantitative conversions with **1D** and **1E** were also obtained within reaction times as low as 10 min (entries 3 and 7). We also obtained quantitative yields for the desired conjugates of eGFP, MBP and mCherry with 1–3 eq. of our *N*-alkyl reagents **1F** and **1G**, although as expected from our small molecule studies (Fig. 3a, c), conjugation with these reagents proceeded slower than that with the *N*-acyl ones, entailing 6 h-long incubations (Fig. 4 and Table 1 entries 10–13 (eGFP), 28–31 (MBP), and 42–43 (mCherry). For these slower *N*-alkyl reagents, we observed the transient accumulation of the

uncyclized intermediates (represented by **B** in Fig. 2a) that ultimately formed the final cyclized bis-heterocyclic adducts (Supplementary Table 6).

Of note, eGFP contains two internal cysteine residues (C51 and C73) that are known to react with thiol-reactive reagents[51], and therefore, can also potentially react with the IBH adducts to form uncyclized thiol conjugates upon acetate elimination (analogous to intermediate **B** in Fig. 2a). Nevertheless, as observed with small molecules, no peaks for these conjugates were observed in the MS analysis of the reactions between the IBH adducts **1A**–**1G** and *N*-Cys eGFP at pH 6.5; instead, peaks for the desired bis-heterocyclic IBH conjugates (Fig. 4, bottom left panel) were observed exclusively. Moreover, the treatment of the *N*-Gly eGFP variant with the IBH adducts **1D** and **1E** did not result in any protein modification (Supplementary Fig. 22). Finally, five of the *N*-Cys labelled protein conjugates, *N*-Cys-eGFP-**1D**, *N*-Cys-eGFP-**1E**, *N*-Cys-eGFP-**1F**, *N*-Cys-MBP-**1D** and *N*-Cys-MBP-**1E**, were digested by trypsin and subjected to LC-MS/MS analysis. The results showed that the eGFP1–7 and MBP1–10 segments carried the IBH modification at the *N*-Cys residue (details in the Supplementary Information section and Supplementary Fig. 23). These observations establish the selectivity of our chemistry for *N*-Cys over internal cysteines of proteins.

Remarkably, the treatment of low/sub-micromolar (down to 100 nM) of *N*-Cys eGFP, MBP and mCherry with merely 1–6 eq. of **1D** and **1E** for 1 h at pH 6.5 gave >90% yields of the desired cyclized conjugates (Table 1, entries 14–21 for eGFP, 32–35 for MBP and 44–46 for mCherry and the corresponding protein conjugate MS spectra are provided in Supplementary Fig. 24). Moreover, quantitative yields were obtained at the lower reaction temperatures of 10 °C (entries 4 and 8 for eGFP; 23, 25 and 27 for MBP; 37, 39 and 41 for mCherry) and 4 °C (entries 5 and 9) as well.

The protein bioconjugates afforded by BHoPAL are stable and retain the structural and functional properties of their unmodified precursor proteins. Indeed, the incubation of 10 μM of our *N*-Cys

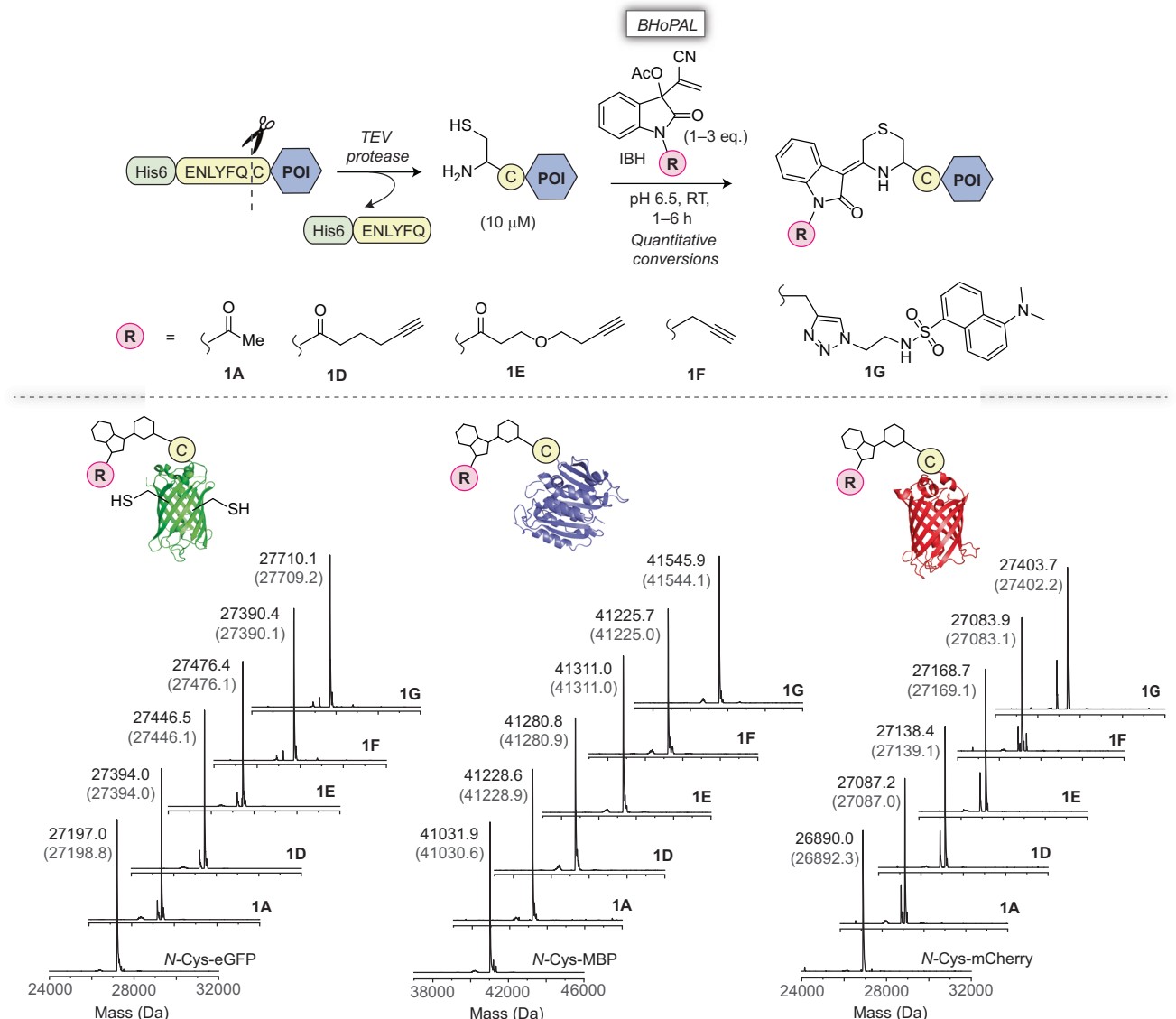

**Fig. 4 | Labelling of *N*-Cys-containing proteins via BHoPAL.** Top: Schematic representation of the labelling workflow involving the generation of *N*-Cys variants of *N*-Cys-eGFP, *N*-Cys-MBP and *N*-Cys-mCherry by employing TEV protease followed by treatment with IBH adducts. The reactions were performed by incubating 10 µM of *N*-Cys-POI with *N*-acyl IBH adducts **1A**, **1D** and **1E** (1.0 eq.) at RT for 1 h in sodium phosphate buffer (50 mM, pH 6.5), or with *N*-alkyl IBH adducts **1F** and **1G** (3.0 eq.) for 6 h under the same conditions. "POI" denotes protein of interest. Bottom: Quantitative conversions were accomplished in all the reactions as depicted by the deconvoluted mass spectra of the protein bioconjugates. The observed masses of the *N*-Cys labelled protein conjugates are shown in black and the corresponding calculated masses are in grey within parenthesis. The MS spectra of the *N*-Cys proteins prior to bioconjugation are depicted in the bottom-most panel. The minor peaks in the mass spectra belong to impurities present in the recombinantly produced proteins. The extended MS traces of unmodified *N*-Cys-POI (control) and *N*-Cys conjugates of IBH adducts are provided in Supplementary Fig. 19.

protein conjugates, eGFP-**1D**, eGFP-**1F**, MBP-**1D** and MBP-**1F**, with excess GSH (5 mM) at 37 °C for 1 day did not alter their MS profiles (Supplementary Fig. 26). Moreover, the CD spectra of our conjugates, eGFP-**1D**, eGFP-**1E**, mCherry-**1D**, mCherry-**1E**, MBP-**1D** and MBP-**1E**, were similar to those of the corresponding unmodified proteins (Supplementary Fig. 27a–c). Additionally, labelling eGFP and mCherry via BHoPAL did not disrupt their intrinsic fluorescent properties, as demonstrated by comparison of the fluorescence spectra of the conjugates eGFP-**1D**, eGFP-**1E**, mCherry-**1D** and mCherry-**1E** with those of the unmodified proteins (Supplementary Fig. 28).

Taken together, these results demonstrate that BHoPAL proceeds rapidly and quantitatively with low reagent concentrations under mild biocompatible conditions without disrupting protein structure and function to form stable protein conjugates.

## BHoPAL in protein mixtures and live cells

Introducing *N*-Cys appended MBP into a mixture of proteins containing eGFP, myoglobin (Mb) and lysozyme (Lyso), followed by treatment with 3.0 eq. of the fluorescent *N*-dansyl IBH adduct **1 G** for 3 h (Fig. 5a) exclusively yielded fluorescently labelled MBP (Fig. 5b, left). In addition to this one-step labelling strategy, we also performed two-step labelling by treating an *N*-Cys MBP-spiked mixture of eGFP, Mb and Lyso with the *N*-alkynyl IBH adduct, **1D** (1.2 eq), followed by a Cu(I)-catalysed azide-alkyne cycloaddition (CuAAC) reaction with dansyl azide (DanN$_3$). This experiment gave similar results as those obtained with the one-step strategy, yielding labelled MBP exclusively (Fig. 5b, right). In addition to in-gel fluorescence, the selective labelling of *N*-Cys-MBP with IBH adducts (**1G** and **1D**) to form the desired IBH-MBP conjugates in a mixture of proteins was convincingly established by

**Table 1 | Efficient *N*-Cys protein bioconjugation at low and sub-micromolar protein and reagent concentrations with BHoPAL**

| Entry | POI (µM) | Reagent | Reagent (eq.) | T (°C) | Time (h) | Conv. (%) |
|---|---|---|---|---|---|---|
| *N*-Cys-eGFP | | | | | | |
| 1 | 10 | **1A** | 1 | 25 | 1 | 100 |
| 2 | 10 | **1D** | 1 | 25 | 1 | 100 |
| 3 | 10 | **1D** | 3 | 25 | 0.16 | 100 |
| 4 | 10 | **1D** | 1 | 10 | 1 | 100 |
| 5 | 10 | **1D** | 1 | 4 | 1 | 100 |
| 6 | 10 | **1E** | 1 | 25 | 1 | 100 |
| 7 | 10 | **1E** | 3 | 25 | 0.16 | 100 |
| 8 | 10 | **1E** | 1 | 10 | 1 | 100 |
| 9 | 10 | **1E** | 1 | 4 | 1 | 100 |
| 10 | 10 | **1F** | 1 | 25 | 6[a] | 100 |
| 11 | 10 | **1F** | 3 | 25 | 6[a] | 100 |
| 12 | 10 | **1G** | 1 | 25 | 6[a] | 66[b] [34] |
| 13 | 10 | **1G** | 3 | 25 | 6[a] | 100 |
| 14 | 5 | **1D** | 1 | 25 | 1 | 100 |
| 15 | 5 | **1E** | 1 | 25 | 1 | 100 |
| 16 | 1 | **1D** | 1 | 25 | 1 | 89 (11) |
| 17 | 1 | **1D** | 3 | 25 | 1 | 98 (2) |
| 18 | 0.5 | **1D** | 1 | 25 | 1 | 83 (17) |
| 19 | 0.5 | **1D** | 3 | 25 | 1 | 96 (4) |
| 20 | 0.1 | **1D** | 3 | 25 | 1 | 71 (29) |
| 21 | 0.1 | **1D** | 6 | 25 | 1 | 100 |
| *N*-Cys-MBP | | | | | | |
| 22 | 10 | **1A** | 1 | 25 | 1 | 100 |
| 23 | 10 | **1A** | 1 | 10 | 1 | 100 |
| 24 | 10 | **1D** | 1 | 25 | 1 | 100 |
| 25 | 10 | **1D** | 1 | 10 | 1 | 100 |
| 26 | 10 | **1E** | 1 | 25 | 1 | 100 |
| 27 | 10 | **1E** | 1 | 10 | 1 | 100 |
| 28 | 10 | **1F** | 1 | 25 | 6[a] | 100 |
| 29 | 10 | **1F** | 3 | 25 | 6[a] | 100 |
| 30 | 10 | **1G** | 1 | 25 | 6[a] | 100 |
| 31 | 10 | **1G** | 3 | 25 | 6[a] | 100 |
| 32 | 5 | **1D** | 1 | 25 | 1 | 100 |
| 33 | 5 | **1E** | 1 | 25 | 1 | 100 |
| 34 | 1 | **1D** | 1 | 25 | 1 | 94 (6) |
| 35 | 0.5 | **1D** | 1 | 25 | 1 | 91 (9) |
| *N*-Cys-mCherry | | | | | | |
| 36 | 10 | **1A** | 1 | 25 | 1 | 100 |
| 37 | 10 | **1A** | 1 | 10 | 1 | 100 |
| 38 | 10 | **1D** | 1 | 25 | 1 | 100 |
| 39 | 10 | **1D** | 1 | 10 | 1 | 100 |
| 40 | 10 | **1E** | 1 | 25 | 1 | 100 |
| 41 | 10 | **1E** | 1 | 10 | 1 | 100 |
| 42 | 10 | **1F** | 3 | 25 | 6[a] | 100 |
| 43 | 10 | **1G** | 3 | 25 | 6[a] | 100 |
| 44 | 5 | **1D** | 1 | 25 | 1 | 100 |
| 45 | 1 | **1D** | 1 | 25 | 1 | 100 |
| 46 | 0.5 | **1D** | 1 | 25 | 1 | 93 (7) |

%Unmodified POI is mentioned in square brackets and %oxidised POI (POI-S-S-POI) in parenthesis.

[a]In addition to 6 h, these experiments were also monitored at 1 h and 3 h time points. The results of these experiments are summarised in Supplementary Table 6 and the deconvoluted MS spectra obtained are provided in Supplementary Fig. 21.

[b]Uncyclized conjugate (12%) and cyclized conjugate (54%); The MS spectra for all the entries are provided in the Supplementary Figs. 19–25: Fig 19 (entries 1, 2, 6, 11, 13, 22, 24, 26, 29, 31, 36, 38, 40, 42 and 43), Fig. 20 (entries 3 and 7), Fig. 24 (entries 14–21, 32–35 and 44–46) and Fig. 25 (entries 4, 5, 8, 9, 23, 25, 27, 37, 39 and 41).

ESI-MS analysis that demonstrated quantitative labelling of *N*-Cys-MBP with our IBH adducts while leaving other proteins in a mixture (Lyso, eGFP and Mb) unmodified (Supplementary Fig. 29).

To explore the potential of BHoPAL for cellular applications, we utilised it for the fluorescent labelling of the cell-surface fusion protein EGFR (epidermal growth factor receptor)-eGFP containing a Cys residue at its *N*-terminus (Fig. 5c)[52]. The transient expression and plasma membrane-localisation of this *N*-Cys EGFR-eGFP fusion protein in HEK293 cells was confirmed by fluorescence imaging of eGFP in the green channel (Fig. 5d). The treatment of these cells with the *N*-alkynyl IBH adduct **1D** at RT for 30 min followed by a live-cell compatible CuAAC reaction[53,54] using the red fluorescent azide, Cy5-N$_3$, imparted red fluorescence only to cells expressing the *N*-Cys EGFR-eGFP protein at the plasma membrane (Fig. 5d and Supplementary Fig. 30). Moreover, untransfected cells (annotated with white arrows in Fig. 5d) not showing green fluorescence for eGFP was not rendered red demonstrating that the red fluorescence was a specific consequence of BHoPAL-mediated protein labelling, and not because of non-specific interactions between the Cy5-N$_3$ dye and other cellular components. This conclusion was reinforced by the results of our experiments wherein cells expressing *N*-Cys EGFR-eGFP were treated directly with Cy5-N$_3$ (without prior treatment with **1D**) under CuAAC conditions that yielded no cellular fluorescence (Supplementary Fig. 31). These experiments demonstrated that BHoPAL is suitable for live-cell protein bioconjugation applications, and proceeds selectively on 1,2-aminothiols not just in vitro but also in live cells.

## BHoPAL at internal 1,2-aminothiols on proteins

A major limitation of 1,2-aminothiol-mediated protein bioconjugation is that it is challenging to introduce this moiety on the protein at sites other than its *N*-terminus. Previously, thiazolidine-appended amino acids have been introduced into proteins at the desired site via the amber codon suppression technology followed by conversion of thiazolidine to the desired 1,2-aminothiol moiety[30,32–34,55]. Although the amber codon suppression approach is powerful, it is technically challenging to execute, often resulting in sub-optimal expression yields (typically <5 mg/L of bacterial culture) of the non-natural amino acid-containing protein[35,36]. Indeed, thiazolidine-containing myoglobin, T4 lysozyme, calmodulin, and ubiquitin produced via this technology were obtained in yields of 2[32], 2[32], 1[32], and 4–5[34,55] mg/L of *Escherichia coli* culture respectively, in contrast to the wild-type versions of these proteins that are have been obtained at much higher yields of 234[56], 50[57], 100[58] and 40[59] mg/L, respectively. Interestingly, an approach involving the incorporation of 1,2-aminothiol reactive nitrile-containing non-canonical amino acids in proteins gave much better expression yields[60].

A technically straightforward and efficient approach for introducing the 1,2-aminothiol moiety at any desired site on proteins will tremendously facilitate the widespread use of BHoPAL and other 1,2-aminothiol labelling-based bioconjugation approaches. We achieved this goal by developing a chemoenzymatic method for introducing the thiazolidine group into any desired site of proteins based on the LAP tag technology that entails the use of the bacterial lipoic acid ligase (LplA) enzyme[61]. LplA catalyses the formation of an isopeptide bond between the ε-amino group of the lysine of its recognition sequence, the LAP tag (GFEIDKVWYDLDA), and the carboxyl group of lipoic acid (LA) (Fig. 6a), and has been previously employed for introducing bioorthogonal functional groups into proteins[61–63].

We envisaged that the structural similarity between thiazolidine (depicted in orange in Fig. 6a) and the dithiolane ring of LA would allow thiazolidine-based LA (TzLA) analogues to serve as substrates for LplA, enabling the introduction of the 1,2-aminothiol moiety (in its

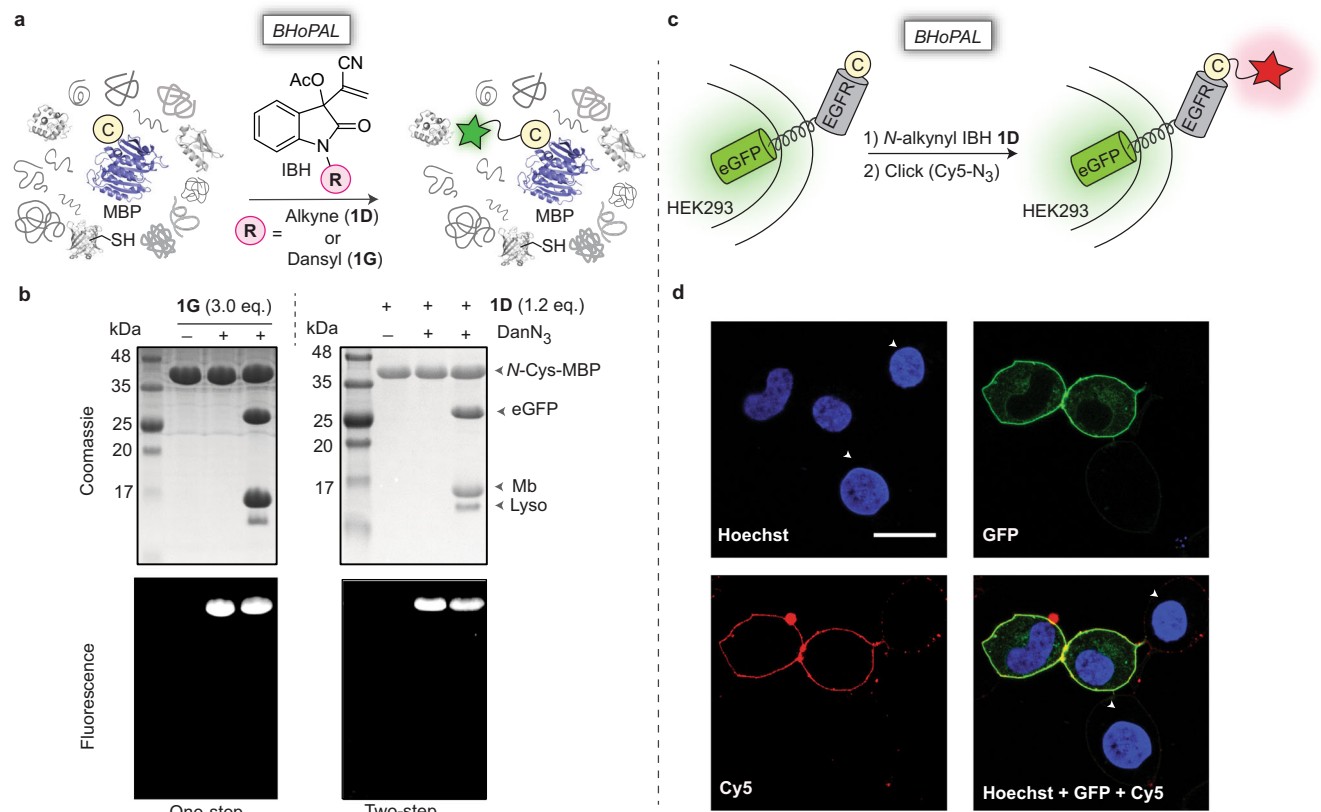

**Fig. 5 | Selective fluorescent labelling of *N*-Cys proteins via BHoPAL in complex in vitro and live-cell conditions. a** A schematic illustration of the experiment on a mixture of proteins. **b** SDS-PAGE analysis of the labelling reaction in a protein pool. An *N*-Cys MBP (50 µM)-spiked mixture of the eGFP (50 µM), Mb (60 µM), and Lyso (50 µM) proteins was treated with 3.0 eq. of the fluorescent *N*-Dansyl IBH adduct **1G** in sodium phosphate buffer at pH 6.5 (one-step bioconjugation, left), or with 1.2 eq. of the *N*-alkynyl IBH adduct **1D** followed by the fluorescent azide, DanN₃ (two-step bioconjugation, right). The ESI-MS analyses for both the labelling reactions are provided in Supplementary Fig. 29. The experiment was performed two times

independently and yielded similar results each time. **c** Schematic depiction of the fluorescent labelling of the *N*-Cys eGFP-EGFR fusion protein in live HEK293 cells. **d** Confocal fluorescence images. HEK293 cells transiently transfected with *N*-Cys EGFR-eGFP were treated with **1D** (50 µM) in sodium phosphate buffer at pH 6.5 for 30 min followed by a CuAAC reaction with the Cy-5 azide dye in PBS for 5 min. Scale bar, 20 µm. Hoechst 33342 was used to stain cellular nuclei. Untransfected cells are highlighted with white arrows. The experiment was performed three times independently and yielded similar results each time.

thiazolidine-protected form) into proteins containing the LAP tag. To test this hypothesis, we employed the W37V variant of LplA that catalyses the ligation of non-natural LA analogues to LAP tag-containing proteins more efficiently than the wild-type enzyme owing to its higher substrate promiscuity[62,64]. Molecular docking studies (detailed in the Supplementary Information section) on the binding of the TzLA analogues, C2–C7Tz, to LplA$^{W37V}$ (Fig. 6b and Supplementary 32a) revealed that the predicted Gibb's free energy (ΔG) of binding increases with increasing carbon chain length with C4 and C5Tz yielding values similar to that for native LA (-61.84, -58.59 and -59.29 Kcal/mol, respectively) in comparison to -67.32 Kcal/mol for C7Tz (Supplementary Table 8). This enhanced energetic stabilisation observed for C7Tz-binding is probably due to three H−bonds (one each with the H149, Y139 and R140 residues of the enzyme) and one salt bridge interaction with H149, in addition to an H−bond between its carboxylic group and the hydroxyl group of the S72 residue of the enzyme (Fig. 6b) that is also observed in the crystal structure of the LA-enzyme complex[65]. Our observation that extending the carbon chain length of the TzLA analogues results in more proficient binding to the enzyme is consistent with the trend previously reported for other LA analogues[66].

To test these docking predictions experimentally, we synthesised the C2–C7Tz analogues (Supplementary Figs. 7 and 8) and incubated them (0.5 mM) with the LAP peptide (250 µM) in the presence of LplA$^{W37V}$ (3 µM) that we produced recombinantly (see Supplementary

Information section for details on enzyme production), and monitored the ligation reaction using HPLC (condition A, Fig. 6c, d). These experiments revealed that, consistent with the predictions of the docking analysis, the longer chain-length analogues were better substrates, with C7Tz forming its LAP conjugate in 92% yield within 1 h, in comparison to C4Tz and C5Tz that gave ligation yields of 31% and 39%, respectively (Fig. 6d, condition A). Moreover, no detectable ligation of C2Tz and C3Tz with the LAP peptide was observed despite incubation time durations as long as 10 h with elevated enzyme and analogue concentrations (10 µM and 1.25 mM, respectively; condition C, Fig. 6d). C4Tz and C5Tz yielded high (>90%) ligation yields only upon 10 h-long incubations with elevated concentrations of the enzyme and the analogues (condition C, Fig. 6d). In contrast, subjecting C7Tz to condition C yielded the desired ligation product in 91% yield in merely 5 min.

We next employed the optimised conditions for LplA-mediated LAP–TzLA conjugation developed above on proteins (Fig. 6e). Specifically, we recombinantly produced a His-tagged MBP variant appended with the LAP tag on its *C*-terminus (details in the Supplementary Information section), and incubated it with 5 eq. of either C7Tz or C5Tz in the presence of LplA$^{W37V}$ (3 µM) for 1 h at pH 7 to obtain the desired thiazolidine-appended MBP in >99% yields as demonstrated by ESI-MS (Fig. 6e, panel II). We obtained a high yield of thiazolidine-appended MBP (C7Tz or C5Tz) using our approach (46 mg from 500 mL of bacterial culture). Subsequently, we quantitatively converted the thiazolidine moiety on the protein into 1,2-aminothiol (Fig. 6e, panel III) by

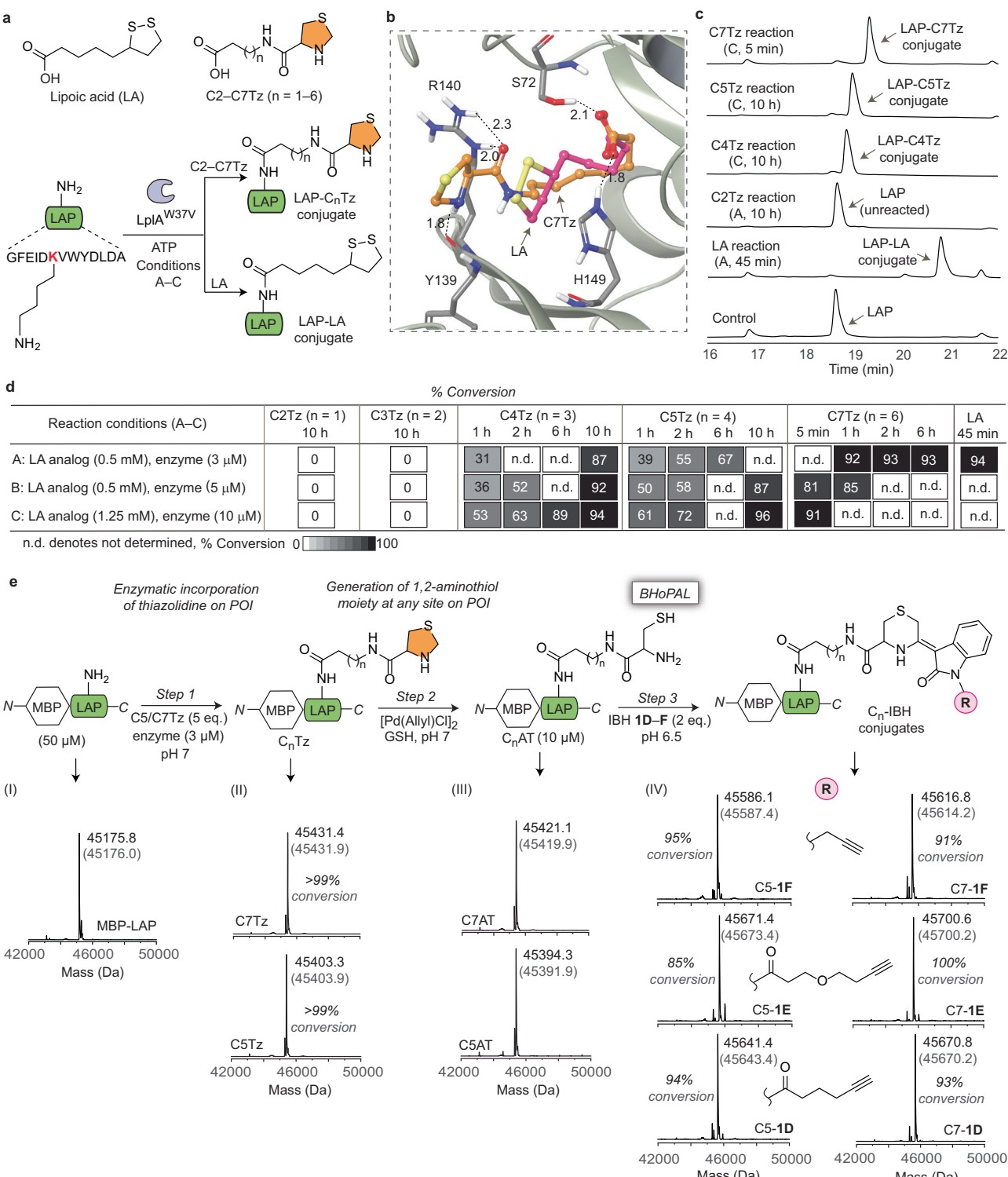

**Fig. 6 | Incorporation of the 1,2-aminothiol group at any site in proteins for BHoPAL. a** Schematic illustration of our strategy for incorporating the 1,2-aminothiol precursor, thiazolidine, on LAP peptide by employing the LplA enzyme. The structures of LA and TzLA analogues (C2–C7Tz) are shown at the top. **b** The binding poses of LA (magenta) and C7Tz (orange) in the binding pocket of LplA$^{W37V}$ generated by the Schrödinger software. The black dotted lines indicate hydrogen bonds between C7Tz and the protein residues. The numbers next to these lines denote H–bond lengths in Å. **c** HPLC analyses of the ligation reactions of LA, C2, C4, C5 and C7Tz with the LAP peptide under conditions "A" and "C" (described in subsection **d** below). Our entire HPLC data on all TzLA analogues are provided in

Supplementary Fig. 34. **d** Table summarising the % yields of ligation obtained with different analogues at various time points under different conditions. All the ligation products were characterised by HRMS (Supplementary Table 9). **e** Top: Scheme for incorporating the 1,2-aminothiol moiety within the MBP-LAP protein followed by BHoPAL. Each of the steps was followed by desalting of the resulting protein conjugate using 10 KDa molecular weight cutoff filters. Bottom: The deconvoluted mass spectra of protein conjugates obtained at each step of the above scheme. The observed masses are shown in black and the corresponding calculated masses are in grey within parenthesis. The XIC and the extended MS traces are provided in Supplementary Fig. 37.

treating it with [Pd(AllylCl)]₂/GSH under previously established reaction conditions[67,68]. Subjecting the resulting 1,2-aminothiol appended MBP variant (10 μM) to BHoPAL by treating it with the IBH reagents **1D**, **1E** or **1F** (2.0 eq.) at RT and pH 6.5 yielded the desired heterocyclic protein bioconjugates (panel IV) in 85–100% yields. Notably, LC-MS/MS analysis of one of the MBP-LAP conjugates, C7-**1F**, yielded *b*- and *y*-ion peaks corresponding to that expected for the labelled LAP tag-containing peptide, validating the modification of the protein at its engineered LAP site (Supplementary Fig. 38). To evaluate the scalability of our protein bioconjugation platform for larger-scale applications, we performed our chemistry on >1 mg MBP-LAP protein and were successful in producing the pure labelled C7-**1D** conjugate in >75% isolated yields (Supplementary Fig. 39).

Moreover, the structural integrity of the MBP-LAP protein remained unperturbed after its modification with our IBH reagents as demonstrated by negligible changes in the CD spectra of two MBP-LAP protein conjugates (C5-**1D** and C7-**1D**) as compared to that for unmodified MBP-LAP (Supplementary Fig. 27d).

These results establish that the use of this LplA-based technology in tandem with BHoPAL provides a facile and high-yielding approach to generate proteins labelled at any desired site, not limited to their *N*-termini.

## Concomitant and Tandem BHoPAL for dual labelling of proteins

Our LAP tag-based approach for installing 1,2-aminothiols at internal sites of proteins described above together with *N*-terminal cysteine installation provides a convenient strategy for the installation of two labels at distinct locations within a protein via BHoPAL. To achieve this goal, we produced an *N*-Cys-MBP variant appended with the LAP tag at its *C*-terminus, and then employed our C7Tz analogue to generate a protein variant (*N*-Cys-C7AT) containing two 1,2-aminothiol moieties (Fig. 7a). To perform dual protein modification in one-step (concomitant BHoPAL), we treated *N*-Cys-C7AT with 5 equivalents of IBH **1E** to obtain the desired dually labelled conjugate, **1E**-C7-**1E**, in 93% conversion as demonstrated by ESI-MS (Fig. 7a).

Next, we investigated the scope of BHoPAL for installing two different labels in proteins by treating *N*-Cys-C7AT with different stoichiometric ratios of **1E** under varying reaction conditions. These experiments demonstrated no preferential labelling of either of the two 1,2-aminothiols (details in Supplementary Table 16). Therefore, to perform selective labelling of each of the two 1,2-aminothiol sites on the protein, we developed a step-wise approach (Tandem BHoPAL). This approach entailed labelling the *N*-terminus of *N*-Cys-MBP-LAP via BHoPAL by treating it with one of our IBH adduct reagents (**1D**), followed by the installation of another 1,2-aminothiol moiety in the LAP tag of the resultant **1D**-MBP-LAP conjugate by employing our C7Tz analog to generate the **1D**-C7AT conjugate. Subsequent treatment with **1E** gave the desired protein conjugate, **1D**-C7-**1E** appended with two different labels at two distinct sites of the protein. Notably, all the steps involved in this workflow proceeded quantitatively as depicted by the ESI-MS (Fig. 7b).

We next evaluated the effects of both mono and dual labelling of MBP via BHoPAL on its ability to bind to its ligand, amylose, by incubating the C7-**1D** (depicted in Fig. 6e) and **1D**-C7-**1E** (above paragraph) conjugates, and the MBP-LAP protein (unlabelled protein) with the amylose-agarose resin (Supplementary Fig. 43). Specifically, these MBP variants were individually incubated with the amylose-agarose resin to allow binding, the bound protein was eluted with maltose-containing elution buffer and % recovery was calculated from the amounts of protein eluted in each case. The results of this assay demonstrated that both the mono and dually labelled conjugates of MBP retained significant binding efficiency with the resin (% recovery obtained for C7-**1D**, **1D**-C7-**1E** and MBP-LAP were 82%, 75% and 97%, respectively).

These results demonstrate that employing BHoPAL together with our LplA-based technology provides a powerful, straightforward, and high-yielding approach for producing dually labelled protein conjugates which is an attractive aspect of our technology as such protein conjugates hold tremendous promise for use in diverse applications including studying protein structure and function, the cellular imaging of proteins, and the fabrication of precision protein therapeutics[69].

## Discussion

Herein, we have disclosed a highly *Z*-selective reaction between IBH adducts and 1,2-aminothiols that rapidly forms a C=C containing bis-heterocyclic scaffold composed of the thiomorpholine and the indolinone cores under physiological conditions. We employed this chemistry to develop the BHoPAL platform that enables the quantitative and site-specific labelling of the 1,2-aminothiol moiety of *N*-Cys residues of proteins. This chemistry proceeds to completion within minutes on low/sub-micromolar protein concentrations in aqueous solutions, in protein mixtures, and in live cells. Furthermore, we have reported a chemoenzymatic technology based on the LplA enzyme that enables the introduction of the 1,2-aminothiol group at any desired site on a protein of interest, rendering 1,2-aminothiol based bioconjugation approaches such as BHoPAL location-agnostic. In combination with BHoPAL, this technology enabled us to produce a variety of protein bioconjugates labelled at the internal sites of proteins. Furthermore, we have extended the scope of our technology by developing a straightforward and high-yielding approach to generate protein conjugates appended with two different labels at two distinct locations. In addition to site-selective protein bioconjugation, the C=C containing bis-heterocyclic core that we have reported herein will enable diverse applications in synthetic and medicinal chemistry.

## Methods

All the reactions were performed in oven-dried glassware under an inert atmosphere of nitrogen or argon. Room temperature (RT) refers to 25 °C. Anhydrous dichloromethane (CH₂Cl₂), acetonitrile (CH₃CN), *N,N*-diisopropylethylamine (DIPEA), and triethylamine (Et₃N) were obtained by distillation over CaH₂ under nitrogen atmosphere. Anhydrous tetrahydrofuran (THF) was prepared by distillation from sodium/benzophenone ketyl. All the reactions were monitored by thin-layer chromatography (TLC) using silica gel precoated plates (0.25 mm thickness). Silica gel of 100–200 mesh was employed for column chromatography. The detailed procedure for the synthesis and the characterisation of all the compounds is provided in the Supplementary Information section. Distilled water was used for reaction work-ups, and ultrapure Type 1 water (Milli-Q water of resistivity 18.2 MΩ.cm at 25 °C) was used for buffer preparations and LC-MS experiments. Degassed buffers were used for in vitro protein bioconjugation.

### Procedure for the synthesis of IBH adducts (1A–G)

**Step 1: General procedure for the synthesis of *N*-substituted isatin.** Procedure 1 (for the synthesis of *N*-alkyl isatin): To a stirred solution of isatin (5 gm, 34 mmoL, 1 eq.) in DMF (40 mL) at 0 °C was added K₂CO₃ (6.5 gm, 47.6 mmoL, 1.5 eq.). After 30 min, alkyl halide (1.2 eq. for methyl iodide or 1.5 eq. for propargyl bromide) was added to the reaction mixture dropwise at 0 °C under a nitrogen atmosphere. The reaction mixture was allowed to warm to RT and stirred for an additional 3 h (for propargyl bromide) or 12 h (for methyl iodide). Water (25 mL) was added, and the reaction mixture was extracted with EtOAc (3 × 50 mL). The organic layers were combined and washed with brine (50 mL), dried with anhydrous Na₂SO₄, and concentrated in vacuo on a rotary evaporator. The residue was purified by column chromatography on silica gel using ethyl acetate/hexane as eluent to give the *N*-alkyl isatin compounds, *N*-propargyl isatin (29%) and *N*-methyl isatin (78%).

Procedure 2 (for the synthesis of *N*-acyl isatin): To a suspension of sodium hydride (1.2 eq.) in anhydrous THF (15 mL) was added a

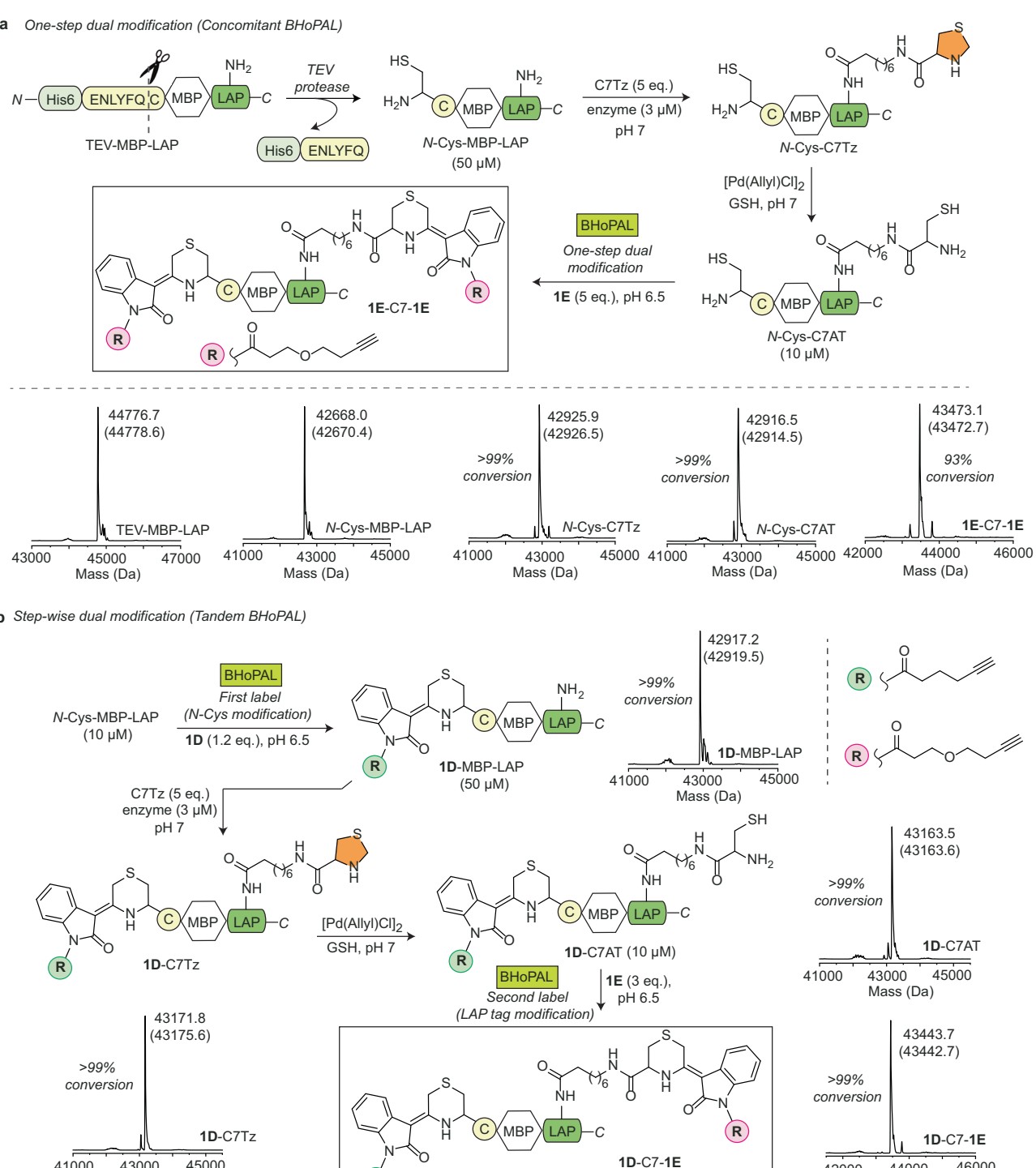

**Fig. 7 | Concomitant and Tandem dual labelling of proteins via BHoPAL. a** One-step dual protein modification (Concomitant BHoPAL). Top: Schematic illustration of our strategy for performing dual labelling of proteins. Bottom: The deconvoluted mass spectra of protein conjugates obtained at each step of the above scheme. **b** Step-wise dual protein modification (Tandem BHoPAL). The deconvoluted mass spectra of protein conjugates obtained at each step are shown next to their structures. The observed masses in all the mass spectra are shown in black and the corresponding calculated masses are in grey within parenthesis. The XIC and the extended MS traces of all the protein conjugates are provided in Supplementary Fig. 42.

solution of isatin (1.9 gm, 13 mmoL, 1 eq.) in THF (25 mL) at 0 °C under nitrogen atmosphere. The resultant dark purple-coloured reaction mixture was stirred for 30 min, and a solution of freshly prepared acyl chlorides (1 eq.) in anhydrous THF (10 mL) was added dropwise to the reaction mixture at 0 °C. The purple colour disappeared upon complete addition and the reaction mixture turned pale yellow. After 15 min of stirring, the reaction was allowed to warm to RT and stirred for an additional 17 h. The reaction was quenched with water (5 mL) and the THF from the reaction mixture was removed in vacuo. The resultant residue was partitioned into EtOAc (50 mL) and water (30 mL). The two layers were separated and the organic layer was washed with brine (30 mL), dried with anhydrous $Na_2SO_4$, and concentrated in vacuo. The residue was purified by column chromatography on silica gel using ethyl acetate/hexane as eluent to give *N*-acyl

isatin compounds in 24–45% isolated yields. The acyl chlorides, 5-hexynoyl chloride and 3-(butynyloxy)propanoyl chloride used in these reactions were synthesised as described in the Supplementary Information section.

**Step 2: General procedure for the Baylis Hillman reaction.** To a stirred solution of *N*-substituted isatin (5.28 mmoL, 1 eq.) in anhydrous THF (15 mL) was added acrylonitrile (4 eq.) and DABCO (1 eq.) at RT under nitrogen atmosphere. The progress of the reaction was monitored by TLC. After stirring for 3.5–14 h, the reaction mixture was concentrated in vacuo, and EtOAc (50 mL) and water (20 mL) were added to the resultant residue. The two layers were separated, and the organic layer was washed with brine (30 mL), dried with anhydrous Na$_2$SO$_4$, and concentrated in vacuo. The residue was subjected to column chromatography on silica gel using ethyl acetate/hexane as eluent to afford *tert*-alcohol Baylis Hillman adducts in 13–91% isolated yields.

**Step 3: General procedure for the *O*-acetylation of *tert*-alcohol Baylis Hillman adducts.** Procedure 1 (for the synthesis of **1 C**): To a stirred and cooled (0 °C) solution of *N*-methyl *tert*-alcohol Baylis Hillman adduct (0.59 gm, 2.7 mmoL, 1 eq.) in anhydrous CH$_2$Cl$_2$ (20 mL) was added acetyl chloride (0.6 mL, 8.2 mmoL, 3 eq.) dropwise under nitrogen atmosphere. Subsequently, K$_2$CO$_3$ (0.761 gm, 5.5 mmoL, 2 eq.) was added and the reaction mixture was stirred for 15 min at 0 °C, and was then allowed to warm to RT and stirred for 43 h. The reaction was quenched with water (20 mL) and the aqueous mixture was extracted with CH$_2$Cl$_2$ (30 mL). The organic layer was washed with brine (20 mL), dried with anhydrous Na$_2$SO$_4$, and concentrated in vacuo. The residue was subjected to column chromatography on silica gel using ethyl acetate/hexane (1:1) as eluent to afford **1 C** as an off-white solid in 18% isolated yield.

Procedure 2 (for the synthesis of **1A, 1B, 1D, 1E, 1F**): To a stirred solution of *N*-substituted *tert*-alcohol Baylis Hillman adducts (3.52 mmoL, 1 eq.) and acetic anhydride (6 eq.) in anhydrous CH$_3$CN (20 mL) was added a solution of Sc(OTf)$_3$ (0.08 eq.) in CH$_3$CN (1 mL) slowly and dropwise at RT under argon atmosphere. After 1 h, the reaction was quenched with saturated NaHCO$_3$ solution in water (10 mL) followed by the addition of EtOAc (30 mL). The two layers were separated, and the organic layer was washed with brine (20 mL), dried with anhydrous Na$_2$SO$_4$, and concentrated in vacuo. The residue was subjected to column chromatography on silica gel using ethyl acetate/hexane as eluent to afford **1A, 1B, 1D, 1E** and **1F** in 47–87% isolated yields.

**Procedure for the synthesis of 1G** (Click reaction of **1F** with DanN$_3$): To a stirred mixture of **1F** (0.105 gm, 0.037 mmoL, 1 eq.) and DanN$_3$ (1 eq.) in CH$_3$CN (10 mL) was added CuI (2 eq.) followed by the addition of DIPEA (3 eq.) at RT. The progress of the reaction was monitored by TLC. After the complete consumption of both the starting materials in about 1 h, the reaction was quenched with 10% citric acid solution in water (10 mL) and the aqueous mixture was extracted with EtOAc (3 × 20 mL). The organic layers were combined and washed with brine (30 mL), dried over anhydrous Na$_2$SO$_4$, and concentrated in vacuo. The resultant residue was subjected to column chromatography on silica gel using ethyl acetate/hexane as eluent to afford **1G** as a brown sticky solid in 57% isolated yield.

**Procedure for the synthesis of thiazolidine-appended lipoic acid analogues (C2–C7Tz)**

Synthesis of C3Tz: To a stirred solution of *N*-boc pentafluorophenyl thiazolidine-4-carboxylate **S8** (0.439 gm, 1.09 mmoL, 1 eq.) in anhydrous CH$_2$Cl$_2$ (20 mL) was added 4-aminobutyric acid (1 eq.) and Et$_3$N (2 eq.) at RT under nitrogen atmosphere. After 5.5 h, the reaction mixture was quenched with 10% citric acid solution in water (20 mL). The resultant two layers were separated and the aqueous layer was

extracted with CH$_2$Cl$_2$ (30 mL × 3). The organic layers were combined and washed with brine (30 mL), dried over anhydrous Na$_2$SO$_4$, and concentrated in vacuo. The residue was subjected to column chromatography on silica gel using CH$_2$Cl$_2$/MeOH (9:1) as eluent to afford C3-boc-Tz in 29% isolated yield. For the deprotection of *N*-boc group, to a stirred solution of C3-boc-Tz (80 mg, 0.254 mmoL, 1eq.) in anhydrous CH$_2$Cl$_2$ (20 mL), TFA (10 eq.) was added dropwise at 0 °C under nitrogen atmosphere. After 15 min, the reaction mixture was allowed to warm to RT and stirred for 5 h. The reaction mixture was concentration in vacuo and the traces of TFA were removed from the residue by evaporating it with toluene (10 mL × 3) under reduced pressure. The resultant residue was subjected to column chromatography on silica gel using CH$_2$Cl$_2$/MeOH (9:1) as eluent to afford zwitter ionic C3Tz as white solid in 58% isolated yield.

Synthesis of C7Tz: To a stirred solution of 8-aminooctanoic acid (0.1 gm, 0.62 mmoL, 1 eq.) in 1 N NaOH solution in water (2 eq.), a solution of *N*-boc pentafluorophenyl thiazolidine-4-carboxylate **S8** (2 eq.) in THF (2 mL) was added dropwise at 0 °C. After 15 min, the reaction mixture was allowed to warm to RT and stirred for 30 min. The reaction mixture was neutralised (pH 7) with 1 N HCl solution in water followed by concentrating the reaction mixture under reduced pressure to remove THF. The resultant neutral aqueous solution was diluted with water (3 mL), and the pH of the solution was adjusted to ~6 with 1 N HCl. Subsequently, the acidic aqueous solution was extracted with EtOAc (30 mL × 3). The organic layers were combined and washed with brine (30 mL), dried over anhydrous Na$_2$SO$_4$, and concentrated in vacuo. The residue was subjected to column chromatography on silica gel using CH$_2$Cl$_2$/MeOH (9:1) as eluent to afford C7-boc-Tz as colourless oil in 26% isolated yield. For the deprotection of *N*-boc group, to a stirred solution of C7-boc-Tz (29 mg, 0.077 mmoL, 1 eq.) in anhydrous CH$_2$Cl$_2$ (10 mL), TFA (12 eq.) was added dropwise at 0 °C under nitrogen atmosphere. After 15 min, the reaction mixture was allowed to warm to RT and stirred for 12 h. The reaction mixture was then concentrated in vacuo and the traces of TFA from the residue were removed by evaporating it with toluene (10 mL × 3) under reduced pressure. The resultant residue was subjected to column chromatography on silica gel using CH$_2$Cl$_2$/MeOH (9:1) as eluent to afford zwitter ionic C7Tz as a white solid in 81% isolated yield.

General procedure for the synthesis of C2Tz, C4Tz and C5Tz: A stirred solution of *N*-boc pentafluorophenyl thiazolidine-4-carboxylate **S8** (0.618 gm, 1.54 mmoL, 1eq.) in anhydrous CH$_2$Cl$_2$ (10 mL) was treated with a solution of *tert*-butyl amino esters (1 eq.) in anhydrous CH$_2$Cl$_2$ (5 mL) at RT under nitrogen atmosphere. Subsequently, Et$_3$N (2 eq.) was added and the reaction mixture was stirred for 15 h followed by concentrating the reaction mixture under reduced pressure. The resultant residue was diluted with EtOAc (40 mL) and 10% citric acid solution in water (20 mL). The two layers were separated, and the aqueous layer was extracted with EtOAc (30 mL × 3). The organic layers were combined and washed with brine (30 mL), dried over anhydrous Na$_2$SO$_4$, and concentrated in vacuo. The residues were subjected to column chromatography on silica gel using hexane/EtOAc as eluent to afford *tert*-butyl esters of C2-boc-Tz, C4-boc-Tz and C5-boc-Tz in 37–68% isolated yields. For the deprotection of *N*-boc and *tert*-butyl ester groups, the resultant compounds (0.482 mmoL, 1 eq.) were dissolved in anhydrous CH$_2$Cl$_2$ (20 mL) and were treated with TFA (12 eq.) at 0 °C under nitrogen atmosphere. After 15 min, the reaction mixtures were allowed to warm to RT, stirred for 8–15 h, and then concentrated in vacuo. Traces of TFA were removed from the resultant residues by evaporating them with toluene (10 mL × 3) under reduced pressure. The residues were subjected to column chromatography on silica gel using CH$_2$Cl$_2$/MeOH (9:1) as eluent to afford C2Tz, C4Tz and C5Tz TFA salts in 55–97% isolated yields. The syntheses of *tert*-butyl 3-aminopropanoate, *tert*-butyl 5-aminopentanoate and *tert*-butyl 6-aminohexanoate employed for the synthesis of C2Tz, C4Tz and

C5Tz, respectively are described in the Supplementary Information section.

## General procedure for the *N*-Cys protein bioconjugation via BHoPAL

*N*-Cys-POI (1 eq.) dissolved in sodium phosphate buffer at pH 7 was treated with TCEP (10 eq. added from a 50 mM stock solution made in water) at RT for 1 h. Subsequently, the protein was desalted using Amicon ultra-0.5 10 K filter, diluted to a concentration of 0.5 mM with degassed Mili-Q water, and used in bioconjugation reactions with IBH adducts. In general, bioconjugation reactions were set up by incubating a solution of TCEP-treated *N*-Cys-POI (5 μL of a 0.5 mM stock in water; final concentration 10 μM) in sodium phosphate buffer at pH 6.5 (220 μL, 50 mM) with IBH adducts **1A/1D/1E** (25 μL, 0.1 mM stock in CH₃CN; final concentration 10 μM, 1 eq.) or **1F/1 G** (25 μL, 0.3 mM stock in CH₃CN, final conc. 30 μM, 3 eq.) at RT. After 1 h (for **1A/1D/1E**) or 3 h (for **1F/1G**) of incubation, the reaction mixtures were desalted using Amicon ultra-0.5 10 K filter and the resultant *N*-Cys-labelled protein conjugates were characterised by ESI-MS analysis (Fig. 4 and Supplementary Fig. 19).

## Procedure for the incorporation of 1,2-aminothiol in MBP-LAP via lipoic acid ligase and its modification via BHoPAL

Step 1: Ligation of C5Tz/C7Tz on the LAP tag of MBP-LAP

C5Tz/C7Tz conjugates of MBP-LAP were prepared by incubating 3 μM of LplA^W37V enzyme (10 μL from 75 μM stock in 1× PBS) with a solution of MBP-LAP (50 μM, 12.5 μL from 1 mM stock in 1× PBS) in sodium phosphate buffer at pH 7 (190 μL, 25 mM) containing Mg(OAc)₂ (5 mM, 12.5 μL from 100 mM stock in reaction buffer) and C5Tz/C7Tz (250 μM, 12.5 μL from 5 mM stock in 50% EtOH/H₂O). Subsequently, the reaction was initiated by the addition of ATP (5 mM, 12.5 μL from 100 mM stock in reaction buffer) and incubated at 37 °C with continuous shaking (300 rpm). After 1 h, the reaction mixture was desalted using an Amicon ultra-0.5 10 K filter and the C5Tz/C7Tz-MBP-LAP conjugates were characterised by ESI-MS analysis (Fig. 6 and Supplementary Fig. 37).

Step 2: Thiazolidine deprotection[68]

C5Tz/C7Tz-MBP-LAP conjugate (50 μM, 20 μL from 0.5 mM stock in water, 1 eq.) was added to sodium phosphate buffer at pH 7 (160 μL, 50 mM) containing [Pd(allyl)Cl]₂ (0.5 mM, 10 μL from 10 mM stock in MeOH, 10 eq.) and GSH (0.5 mM, 10 μL from 10 mM stock in water, 10 eq.). The reaction mixture was incubated at 37 °C with continuous shaking (150 rpm) and after 1 h, the reaction was quenched by adding DTT (20 μL from a 200 mM stock in water). After 30 min of incubation at 37 °C, the reaction mixture was desalted by using an Amicon ultra-0.5 10 K filter and the resultant 1,2-aminothiol-appended conjugates, C5AT and C7AT were characterised by ESI-MS (Fig. 6 and Supplementary Fig. 37).

Step 3: Labelling of MBP-LAP via BHoPAL

A solution of C5AT/C7AT (5 μL from 0.5 mM stock in water, 1 eq.) in sodium phosphate buffer at pH 6.5 (220 μL, 50 mM) was treated with IBH adduct **1D/1E/1F** (20 μM, 25 μL from 0.2 mM stock in CH₃CN, 2 eq.) at RT. After 5 h of incubation, the reaction mixture was desalted by using an Amicon ultra-0.5 10 K filter and the resultant bis-heterocyclic protein conjugates (C5-**1D**, C5-**1E**, C5-**1F**, C7-**1D**, C7-**1E**, C7-**1F**) were characterised by ESI-MS (Fig. 6 and Supplementary Fig. 37).

## Reporting summary

Further information on research design is available in the Nature Portfolio Reporting Summary linked to this article.

## Data availability

All data supporting the results and conclusions are available in the supplementary information section. These data include synthetic procedures, compound characterisation, HPLC studies on reaction kinetics and conjugate stability, protein expression and purification procedures, protein bioconjugation optimisation, MS characterisation of protein bioconjugates, CD characterisation of protein bioconjugates, functional studies on protein bioconjugates, cellular imaging studies, and docking studies. Source data containing raw and uncropped gels and confocal images of the main text and supplementary figures are provided in this paper. The *X*-ray crystallographic coordinates for structures reported in this study have been deposited at the Cambridge Crystallographic Data Centre (CCDC), under deposition numbers 2259911 and 2259910. These data can be obtained free of charge from The Cambridge Crystallographic Data Centre via www.ccdc.cam.ac.uk/data_request/cif. Source data are provided in this paper.

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

## Acknowledgements

The authors thank Prof. Oliver Seitz for his kind gift of the plasmid for expressing Cys-E3-EGFR-eGFP, and Prof. Sunando Datta for the amylose-agarose resin. This work was supported by funds from IISER Bhopal and SERB POWER grant SPG/2021/004151 awarded to D.K. The imaging data shown was obtained from the imaging facility at IISER Bhopal supported by DST-FIST. M.H.M. thanks CSIR for the fellowship. S.P. and C.S. thank IISER Bhopal for the fellowship.

## Author contributions

D.K. conceived the project. M.H.M. synthesised the compounds and performed small molecule conjugation and kinetic analyses. S.P. cloned and produced recombinant proteins, and performed in vitro and in cellulo-protein bioconjugation, and LAP tag peptide modification experiments. C.S. performed the molecular docking experiments. D.K., M.H.M., S.P. and C.S. wrote the first draft of the manuscript. D.K. edited the manuscript and wrote the final draft.

## Competing interests

The authors declare no competing interests.
