## [Peer Review File · Nature Communications]

REVIEWER COMMENTS

Reviewer #1 (Remarks to the Author):

Location-agnostic site-specific protein bioconjugation via Baylis-Hillman Adducts

Kalia and co-workers describe a new method for N-terminal cysteine (N-Cys) bioconjugation by utilizing isatin-derived Baylis-Hillman (IBH) adducts, where the new chemistry is referred to as 'BHoPAL' (Baylis-Hillman adduct orchestrated protein aminothiols labelling). The authors carefully characterize this new chemistry, revealing a high reaction rate ($k > 10^3 \text{ M}^{-1}\text{s}^{-1}$) alongside chemoselectivity (towards 1,2-aminothiols) and Z-regioselectivity of the formed adduct (due to hydrogen bonding). Protein labeling was demonstrated with three N-Cys-proteins (eGFP, MBP and m-Cherry) via expression of the corresponding TEV-protease substrates on the N-termini. Subsequently, labeling was demonstrated in vitro (one-step and two-step with CuAAC) and under live-cell conditions (two-step with CuAAC).

In addition, the authors report a new method for enzymatic installation of the 1,2-aminothiol modality using lipoic acid ligase. Using a mutant of the enzyme which has more substrate promiscuity (Lp1Aw37v), the authors screen for an optimal carbon length of a thiazolidine (Tz) derivative of lipoic acid (LA) to achieve ligation. These Tz derivatives can be rapidly ligated, followed by palladium/GSH deprotection to obtain the 1,2-aminothiol, which is amenable for 'BHoPAL' (or any other 1,2-aminothiol-selective chemistry).

The development of the BHoPAL bioconjugation technique, and its synergy with the chemo-enzymatic strategy described are of interest to the readership. I recommend publication of the manuscript in Nature Communications upon addressing the concerns/suggestions mentioned below:

Major

1. While Figure 6 presents impressive data, I would encourage a more elaborate discussion on the outlook of this overall method (see ideas below) as a more appropriate ending of the manuscript.
 - In the manuscript, the LAP tag was introduced on the C-terminus of MBP to enable the lipoic acid ligase strategy (Fig. 6). Did the authors also attempt usage of their lipoic acid ligase strategy when the LAP tag was introduced at other sites within proteins? Or is the strategy limited to N- and C-termini?
 - What is the prospect of this method for multivalent protein modification? Is it feasible, or do the authors suspect limitations (for instance due to the overall hydrophobicity of the IBH-modification)?
 - While not necessary for the method development, an experiment with a functionalized bioconjugate might be illustrative. For instance, in addition to the shown CD spectra, demonstrating the retained functionality of the modified MBP-LAP protein conjugate.
2. Figure 6e: Intermediate purification steps are necessary to enable successive execution of steps 1 (lipoic acid ligase), 2 (thiazolidine \rightarrow 1,2-aminothiol) and 3 (IBH adduct treatment). These purifications were executed using Amicon 10K filters (according to the SI). I think this should be more clearly described in the manuscript itself. In addition, rather than only reporting conversions based on MS data, I would like to see more context on the overall yield of the entire process on a preparative scale ($>1 \text{ mg?}$).

3. Concerning the mechanism of the overall BHoPAL method: what happens exactly when thiols are employed that lack a nearby amino group? Is it merely reversibility of the Thio-Michael addition (preventing the acetate group from ever leaving)? This topic was also mentioned on the protein level for eGFP (internal competition), but I believe that a small molecule experiment might be more informative:

Figure 3b / S2: I would suggest to add a competition experiments with an excess of N-acetyl cysteine (NAC) or GSH at pH 8, as this provides the highest probability of identifying potential side products occurring when the Aza-Michael addition does not occur. Specifically, this could assist in providing more detail to the reaction mechanism.

4. Please collate all supporting Figures, Schemes and Tables into separate groups at the beginning of the SI. It is currently rather difficult to find specific supporting figures/schemes/tables which are mentioned in the main text.

Minor

5. Figure 3a: If the reaction is complete within 2 minutes, HPLC analysis (taking up to 30 minutes, judging by the reference for 1A-C) may not be ideal unless a quenching step occurs before injection of the sample (this is not specified by the authors).

6. Figure 3 / S4: For clarity, please indicate the method of normalization to translate the raw absorbance data into the conversion (%) and rates.

7. When comparing the reaction rate of the bioconjugation reaction compared to other N-terminal cysteine-targeted methods, the authors point out that thiazolidino boronate (TzB) conjugates lack stability. However, the authors themselves cited a follow-up paper (ref 27) which reports an acyl transfer to solve this (rate constant $\sim 5000 \text{ M}^{-1}\text{s}^{-1}$ followed by a slower acyl transfer which is optimal below pH 7). I believe that omitting this nuance in the discussion of the reaction rate should be reconsidered.

8. Figure 5b (left / right): it's not immediately clear what the differences in conditions are between the two lanes that are treated with 1G or the two lanes that are treated with 1D and DanN3. I also could not find this information in the legend of Figure 5.

9. Figure 5d: could the authors also discuss whether they attempted one-step labelling with a fluorescently conjugated IBH reagent, as excluding the copper click would be favourable. I could imagine that solubility of the reagent is the limiting factor?

Reviewer #2 (Remarks to the Author):

The manuscript entitled "Location-agnostic site-specific protein bioconjugation via Baylis-Hillman adducts" presents a novel and rapid diastereoselective reaction between isatin-derived Baylis-Hillman adducts and 1,2-aminothiols to form a stable bis-heterocyclic scaffold. The authors utilize this reaction for protein bioconjugation under physiological conditions and demonstrate its application in live-cell conditions. Additionally, they introduce a lipoic acid ligase-based technology to install the 1,2-aminothiol moiety at any desired site within proteins, overcoming the limitations of existing methods and enabling labeling at internal sites of proteins having 2-aminothiol moiety as its side chain. The study holds

significant promise for chemical biology and drug development applications. The manuscript is well written for a broader audience scope and the analytical data's (MS & NMR of the small molecules) are accurate.

Strengths:

1. The manuscript addresses a critical challenge in protein bioconjugation by introducing an ultrafast and highly selective reaction under physiological conditions. This advancement reduces the need for elevated reagent concentrations and enhances the stability of protein bioconjugates.
2. The development of the BHoPAL platform enables quantitative and site-specific labeling of the 1,2-aminothiol moiety of N-terminal cysteines (N-Cys) of proteins. The method exhibits efficient protein labeling within minutes, including live-cell applications.
3. The incorporation of a lipoic acid ligase-based technology expands the applicability of 1,2-aminothiol-based bioconjugation approaches beyond N-terminal labeling, making it location-agnostic and facilitating labeling at internal sites of proteins.
4. The novel bis-heterocyclic scaffold comprising thiomorpholine and indolinone cores presents potential applications in synthetic and medicinal chemistry, opening avenues for further exploration.

Major revisions:

1. The authors should provide a clear and comprehensive HPLC diagram, preferably a full run, and accurately depict the unreacted amino acids peak in HPLC traces in Fig S3a. The current representation lacks visibility of unreacted amino acids, except tryptophan and phenylalanine, making it difficult to assess the extent of the reaction.
2. The inconsistency in the TEV protease cleavage sequence needs to be addressed. In page 7, the sequence is written as ENLYGQC, whereas in Fig 4 and the Supplementary Information (SI), it is written as ENLYFQC.
3. The authors should provide MS/MS spectra for either all or some of the MBH adducts modified POI in Fig S9 to further validate the exclusive ligation sites at the N-terminus. This additional data will strengthen the evidence for site-specific labeling.
4. The reviewer would like to know the selectivity of the developed methodology between a N-terminal cysteine and a aminothiol moiety at the side chain as introduced via a lipoic acid ligase-based technology. They may choose to perform the selectivity demonstration experiment either on a N-Cys-MBP-Lap-C or may be on a synthetic peptide containing both the moiety. That will give a more refined idea about the selectivity of the platform between N-terminal & side chain 2-aminothiol moiety.
5. One of the major strengths of this method is that it's ability to modify N-Cys proteins even at sub-micromolar concentration (upto 100 nm protein concentration). Did authors try the method to detect naturally occurring N-Cys containing active sites (either known sites or newly discovered sites) in the human proteome?
6. The authors should provide MS/MS data along with the intact mass [C5-1D, C5-1E, C5-1F] & [C7-1D, C7-1E, C7-1F] (any one of them is enough) to further confirm the ligation at the specified position.

7. The authors should remove the word “diastereoselective” (Fig 1a) and “stereospecific” from the manuscript. Although it is forming exclusively Z-isomer after the formation of bis-heterocyclic core, the reaction is not diastereoselective in its actual form (since no new stereocenters are being generated after the reaction). They may call it (Z)-selective if they want.

Overall, the manuscript introduces an innovative approach for location-agnostic site-specific protein bioconjugation, effectively overcoming existing limitations. The demonstrated rapid reaction kinetics and broad applicability make it highly valuable for chemical biology and drug development. Addressing the major corrections mentioned above will enhance the clarity and scientific rigor of the manuscript. I recommend acceptance of the paper for publication in Nature Communications after the abovementioned revisions.

Reviewer #3 (Remarks to the Author):

In this manuscript, a new strategy for the modification of N-terminal Cys residues and 1,2-aminothiol motifs is disclosed using functionalized, isatin-derived Baylis-Hillman adducts. Proof of concept for the reaction of cysteamine with the IBH adducts is explored, confirming reactivity in aqueous media, with reaction rate largely dependent on the pH of the reaction mixture. Chemoselectivity is analyzed through a robustness study in the presence of various amino acids as additives. The reaction is translated to protein examples bearing N-terminal Cys residues exposed via a TEV protease cleavage site. This reaction proceeds efficiently, with stoichiometric or small excesses of IBH reagent, and importantly, is viable at low concentrations (e.g. sub-micromolar range). The rate of reaction is responsive to the nature of the IBH “R” group, with acyl derivatives reacting more rapidly than the corresponding alkyl variants. Labeling of a protein mixture and of live cells expressing N-Cys labelled eGFP is demonstrated in an efficient manner. Finally, a strategy for introducing 1,2-amino thiols into alternative sites within the protein (apart from the N-terminus) is also established which cleverly utilizes the lipoic acid ligase machinery. Optimization of lipoic acid variants featuring a thiazolidine motif is carried out to maximize the yield and rate of incorporation into peptides using the enzyme LplA. Unmasking of the thiazolidine reveals the 1,2-aminothiol motif primed for conjugation. Apart from the IBH chemistry illustrated here, this approach serves to extend the scope and applicability of bioconjugation strategies developed for N-terminal Cys residues.

In my opinion, the work is well-executed, including a detailed supporting information file, and will be of broad interest to the community. I therefore support publication in Nature Communications. I would suggest the following revisions prior to publication.

1. The slow reaction kinetics of existing N-terminal Cys modification strategies is highlighted in the introduction as a major drawback of existing strategies. As a term “slow,” is quite relative. Comparative rate constants for some of the alternative strategies are later discussed (p. 5), but given variability in the measurement and reporting of rate constants from different publications, it would be interesting to know how the IBH reaction fares in a direct competition study, e.g. if cysteamine is treated simultaneously with IBH adduct 1A and an alternative electrophile (e.g. a model thioester – to mimic NCL, with CBT, or cyclopropanone), how much of the desired product is formed? Alternatively, the authors could attempt one of the other approaches under analogous conditions to the IBH chemistry and track reaction progress via HPLC.

2. Stability in the presence of external thiols (e.g. glutathione) is evaluated. What happens to the product in the presence of an exogenous 1,2-aminothiol (e.g. cysteamine)? Is there any reversibility observed?

3. In the SI, the stereochemistry of Cys in S2A-Cys should be checked (was this L or D-Cys?); the D-isomer is drawn here, but presumably the more abundant L-amino acid would have been used? The same error is repeated in Scheme S4 (conversion to the activated ester of the thiazolidine) and throughout other thiazolidine examples in the SI.

4. In Table S6 in the SI, uncyclized IBH-POI adduct (B%) is quantified at different time points. However, this is not mentioned in the main text. While this does not necessarily indicate any lack of selectivity for N-terminal as opposed to internal Cys residues (as the N-terminal Gly eGFP, for example, showed no modification), it would still be valuable to comment on this point in the text. Is this initial thiol reaction step reversible, or does the appended amine in the 1,2-aminothiol motif increase the reactivity of the thiol in the first step of the reaction, therefore leading to the formation of the uncyclized IBH-POI conjugate only at the N-terminus?

Point-wise response to reviewer's comments (reviewer comments in black and our responses in red).

We thank the reviewers for their insightful comments on our work. Addressing these comments has improved the quality of our manuscript.

Reviewer #1 (Remarks to the Author):

Location-agnostic site-specific protein bioconjugation via Baylis-Hillman Adducts
Kalia and co-workers describe a new method for N-terminal cysteine (N-Cys) bioconjugation by utilizing isatin-derived Baylis-Hillman (IBH) adducts, where the new chemistry is referred to as 'BHoPAL' (Baylis-Hillman adduct orchestrated protein aminothiols labelling). The authors carefully characterize this new chemistry, revealing a high reaction rate ($k > 10^3 \text{ M}^{-1} \text{ s}^{-1}$) alongside chemoselectivity (towards 1,2-aminothiols) and Z-regioselectivity of the formed adduct (due to hydrogen bonding). Protein labeling was demonstrated with three N-Cys-proteins (eGFP, MBP and m-Cherry) via expression of the corresponding TEV-protease substrates on the N-termini. Subsequently, labeling was demonstrated in vitro (one-step and two-step with CuAAC) and under live-cell conditions (two-step with CuAAC). In addition, the authors report a new method for enzymatic installation of the 1,2-aminothiol modality using lipoic acid ligase. Using a mutant of the enzyme which has more substrate promiscuity (Lp1Aw37v), the authors screen for an optimal carbon length of a thiazolidine (Tz) derivative of lipoic acid (LA) to achieve ligation. These Tz derivatives can be rapidly ligated, followed by palladium/GSH deprotection to obtain the 1,2-aminothiol, which is amenable for 'BHoPAL' (or any other 1,2-aminothiol-selective chemistry). The development of the BHoPAL bioconjugation technique, and its synergy with the chemo-enzymatic strategy described are of interest to the readership. I recommend publication of the manuscript in Nature Communications upon addressing the concerns/suggestions mentioned below:

Our response: We thank the reviewer for recommending our work for publication in *Nature Communications*, and have addressed all of their concerns/suggestions as described below.

Major

1. While Figure 6 presents impressive data, I would encourage a more elaborate discussion on the outlook of this overall method (see ideas below) as a more appropriate ending of the manuscript.

- In the manuscript, the LAP tag was introduced on the C-terminus of MBP to enable the lipoic acid ligase strategy (Fig. 6). Did the authors also attempt usage of their lipoic acid ligase strategy when the LAP tag was introduced at other sites within proteins? Or is the strategy limited to N- and C-termini?

Our response: We did not attempt to introduce the LAP tag at internal sites of proteins but believe that this will not be a limitation. There are reports in the literature that demonstrate efficient lipoic acid ligase-mediated of protein labeling of the LAP tags present at the internal sites. For example, Kaar and co-workers have demonstrated lipoic acid ligase-mediated conjugation of 10-azidodecanoic acid at two internal positions as well as on the N-terminus of GFP (*Bioconjugate Chem.* **2015**, 26, 6, 1104–1112). Considering that the introduction of LAP tags and their use for protein labelling at internal sites of proteins is well-established, we

believe that our method can be utilized to label LAP tags of the protein introduced at any site of a protein.

- What is the prospect of this method for multivalent protein modification? Is it feasible, or do the authors suspect limitations (for instance due to the overall hydrophobicity of the IBH-modification)?

Our response: We thank the reviewer for suggesting this idea to us and we have addressed this point and included the new data generated in the revised version of the manuscript. To explore this interesting idea, we successfully generated dually-labelled protein conjugates efficiently and in high yields via BHoPAL. These results have been introduced as a new section in the revised version of the main text as “Concomitant and tandem dual labelling of proteins via BHoPAL” on page 14 of the main text, and as a new figure (Figure 7). All the procedures, supporting figures and tables corresponding to this work have been included in the revised version of the supporting file on pages 103–110.

Briefly, to achieve this goal, we produced an MBP variant appended with two 1,2-aminothiol moieties (one at the *N*-terminus and other on the LAP tag) and treated it with 5 equivalents of our IBH reagent, **1E**, to obtain the desired dually-labelled conjugate in 93% conversion (Concomitant BHoPAL). Additionally, we also developed a step-wise approach (Tandem BHoPAL) to quantitatively generate dually-labelled protein bioconjugates appended with different labels at two distinct specific sites on a single protein molecule.

With respect to the reviewer’s point about complications due to the hydrophobicity of the reagents, we did not face any problems due to reagent hydrophobicity probably because we did not need to use excess reagents, and have utilized merely 1.2–5 equivalents of IBH adducts **1D** and **1E** to generate dually-labelled protein conjugates in high yield.

- While not necessary for the method development, an experiment with a functionalized bioconjugate might be illustrative. For instance, in addition to the shown CD spectra, demonstrating the retained functionality of the modified MBP-LAP protein conjugate.

Our response: We have addressed this comment in our revised submission by evaluating the effects of both mono and dual labelling of MBP via BHoPAL on its ability to bind to its ligand, amylose. We perform this assay by evaluating the binding efficiency of unlabelled, mono-labelled and dually-labelled MBP with the amylose-agarose resin. The results of this assay demonstrated that these conjugates retained significant binding efficiency with the resin (details are provided in the revised main text on page 14 and in Fig. S36 on pages 110–111 of the SI section).

2. Figure 6e: Intermediate purification steps are necessary to enable successive execution of steps 1 (lipoic acid ligase), 2 (thiozolidine → 1,2-aminothiol) and 3 (IBH adduct treatment). These purifications were executed using Amicon 10K filters (according to the SI). I think this should be more clearly described in the manuscript itself. In addition, rather than only reporting conversions based on MS data, I would like to see more context on the overall yield of the entire process on a preparative scale (>1 mg?).

Our response: We have now added a separate sentence in the main text (in the legend of Figure 6, page 12) to clearly mention the desalting steps performed. Also, as per the reviewer’s suggestion, we have evaluated the scalability of our protein bioconjugation platform and performed our chemistry on >1 mg MBP-LAP protein and were successful in producing the pure labelled C7-**1D** conjugate in >75% isolated yields over three steps (a

schematic representation of the experiment is introduced in the revised SI section as Fig. S32 on page 99, and these results have been discussed on page 12 of the revised main text).

3. Concerning the mechanism of the overall BHoPAL method: what happens exactly when thiols are employed that lack a nearby amino group? Is it merely reversibility of the Thio-Michael addition (preventing the acetate group from ever leaving)? This topic was also mentioned on the protein level for eGFP (internal competition), but I believe that a small molecule experiment might be more informative:

Figure 3b / S2: I would suggest to add a competition experiments with an excess of N-acetyl cysteine (NAC) or GSH at pH 8, as this provides the highest probability of identifying potential side products occurring when the Aza-Michael addition does not occur. Specifically, this could assist in providing more detail to the reaction mechanism.

Our response: We have observed that the reactions between thiols such as internal cysteines of proteins that are not appended with an adjacent amine group and the Michael acceptor moiety of IBH adducts are not reversible. These thiols when treated with IBH adducts undergo addition followed by the elimination of the acetate group to form a compound akin to **B** in Figure 2 of the main text. However, we noticed that the reactivity of the 1,2-aminothiols towards IBH adducts is substantially higher than alkyl thiols lacking an adjacent amine group. In the revised version, we have addressed this issue by reporting studies comparing the reactivity of the alkane thiols NAC and 1-hexanethiol with our IBH adduct **1A**, with that between cysteamine and **1A**. We observed excellent selectivity of **1A** for reacting with cysteamine as compared to the other two alkane thiols (>90% at pH 8 and as high as 97% at pH 6.5). We have reported these experiments and discussed these results in the revised main text on page 5–6 and in Fig. S4 of the revised SI section on page 40–41. These results suggest that the considerably higher reactivity of 1,2-aminothiols (as compared to other thiols) enables the exclusive formation of its conjugates when equimolar amounts/slight excess of the IBH adducts are employed. The 1,2-aminothiol selectivity of our chemistry is therefore, not due to the reversibility of the reactions of IBH adducts with other thiols, but rather, due to the considerably higher propensity of 1,2-aminothiols for reacting with these adducts.

4. Please collate all supporting Figures, Schemes and Tables into separate groups at the beginning of the SI. It is currently rather difficult to find specific supporting figures/schemes/tables which are mentioned in the main text.

Our response: At the beginning of the revised SI section, we have introduced a Table of Contents depicting the page numbers on which the supporting figures, schemes and tables appear in the SI section.

Minor

5. Figure 3a: If the reaction is complete within 2 minutes, HPLC analysis (taking up to 30 minutes, judging by the reference for 1A-C) may not be ideal unless a quenching step occurs before injection of the sample (this is not specified by the authors).

Our response: To address this point, we now report quenching experiments wherein we add HCl to the reactions to quench the reaction (due to low pH) before subjecting them to HPLC analyses. Specifically, in these experiments, aliquots from the reaction mixture were withdrawn at different time points and quenched with 1M HCl before subjecting them to HPLC analysis. The results of these experiments demonstrate no change in the reaction

profiles as compared to those performed without quenching that are reported in Fig. 3a of the main text. These HPLC traces obtained in the quenching experiment are included on the page 36 of the SI.

6. Figure 3 / S4: For clarity, please indicate the method of normalization to translate the raw absorbance data into the conversion (%) and rates.

Our response: The information is added in the procedure section of the SI on page 42.

7. When comparing the reaction rate of the bioconjugation reaction compared to other N-terminal cysteine-targeted methods, the authors point out that thiazolidino boronate (TzB) conjugates lack stability. However, the authors themselves cited a follow-up paper (ref 27) which reports an acyl transfer to solve this (rate constant $\sim 5000 \text{ M}^{-1}\text{s}^{-1}$ followed by a slower acyl transfer which is optimal below pH 7). I believe that omitting this nuance in the discussion of the reaction rate should be reconsidered.

Our response: We have rewritten this sentence on page 2 of the main text for better clarity.

8. Figure 5b (left / right): it's not immediately clear what the differences in conditions are between the two lanes that are treated with 1G or the two lanes that are treated with 1D and DanN3. I also could not find this information in the legend of Figure 5.

Our response: Whereas the conjugation with **1G** (a dansyl IBH adduct) was performed in one step (by using 3 eq. of **1G**), the reaction with **1D** (an alkynyl IBH adduct) was performed in two steps (first performing the BHoPAL chemistry and then a click reaction with DanN₃). We have highlighted this text in the Figure 5 legend in red.

9. Figure 5d: could the authors also discuss whether they attempted one-step labelling with a fluorescently conjugated IBH reagent, as excluding the copper click would be favourable. I could imagine that solubility of the reagent is the limiting factor?

For *in vitro* bioconjugation applications, we have developed a fluorescent IBH adduct (**1G**) and have used it for one-step protein labelling via BHoPAL (Figure 5b). However, we have not yet explored one-step BHoPAL for *in cellulo* bioconjugation, and are currently developing the requisite IBH adducts in our laboratory.

Reviewer #2 (Remarks to the Author):

The manuscript entitled "Location-agnostic site-specific protein bioconjugation via Baylis-Hillman adducts" presents a novel and rapid diastereoselective reaction between isatin-derived Baylis-Hillman adducts and 1,2-aminothiols to form a stable bis-heterocyclic scaffold. The authors utilize this reaction for protein bioconjugation under physiological conditions and demonstrate its application in live-cell conditions. Additionally, they introduce a lipoic acid ligase-based technology to install the 1,2-aminothiol moiety at any desired site within proteins, overcoming the limitations of existing methods and enabling labeling at internal sites of proteins having 2-aminothiol moiety as its side chain. The study holds significant promise for chemical biology and drug development applications. The manuscript is well written for a broader audience scope and the analytical data's (MS & NMR of the small molecules) are accurate.

Strengths:

1. The manuscript addresses a critical challenge in protein bioconjugation by introducing an ultrafast and highly selective reaction under physiological conditions. This advancement reduces the need for elevated reagent concentrations and enhances the stability of protein bioconjugates.
2. The development of the BHoPAL platform enables quantitative and site-specific labeling of the 1,2-aminothiol moiety of N-terminal cysteines (N-Cys) of proteins. The method exhibits efficient protein labeling within minutes, including live-cell applications.
3. The incorporation of a lipoic acid ligase-based technology expands the applicability of 1,2-aminothiol-based bioconjugation approaches beyond N-terminal labeling, making it location-agnostic and facilitating labeling at internal sites of proteins.
4. The novel bis-heterocyclic scaffold comprising thiomorpholine and indolinone cores presents potential applications in synthetic and medicinal chemistry, opening avenues for further exploration.

Our response: We thank the reviewer for praising our work.

Major revisions:

1. The authors should provide a clear and comprehensive HPLC diagram, preferably a full run, and accurately depict the unreacted amino acids peak in HPLC traces in Fig S3a. The current representation lacks visibility of unreacted amino acids, except tryptophan and phenylalanine, making it difficult to assess the extent of the reaction.

Our response: The full HPLC traces have now been added in the revised SI section (Fig. S3 on page 39).

2. The inconsistency in the TEV protease cleavage sequence needs to be addressed. In page 7, the sequence is written as ENLYGQC, whereas in Fig 4 and the Supplementary Information (SI), it is written as ENLYFQC.

Our response: We thank the reviewer for pointing out this inadvertent error on our part. The correct sequence is ENLYFQC, and we have corrected this error in the revised main text.

3. The authors should provide MS/MS spectra for either all or some of the MBH adducts modified POI in Fig S9 to further validate the exclusive ligation sites at the N-terminus. This additional data will strengthen the evidence for site-specific labeling.

Our response: We have addressed this comment by performing trypsin digestion followed by MS/MS analyses of five of our N-Cys labelled protein conjugates: *N-Cys-eGFP-1D*, *N-Cys-eGFP-1E*, *N-Cys-eGFP-1F*, *N-Cys-MBP-1D* and *N-Cys-MBP-1E*. The results confirmed one IBH modification at the N-Cys residue for each of these conjugates. The discussion pertaining to the MS/MS experiment is introduced on page 7–8 of the revised main text. The LC-MS/MS analyses of the digested protein conjugates is included in the Fig. S16 on page 68–70 of the revised SI section.

4. The reviewer would like to know the selectivity of the developed methodology between a N-terminal cysteine and a aminothiols moiety at the side chain as introduced via a lipoic acid ligase-based technology. They may choose to perform the selectivity demonstration experiment either on a N-Cys-MBP-Lap-C or may be on a synthetic peptide containing both the moiety. That will give a more refined idea about the selectivity of the platform between N-terminal & side chain 2-aminothiol moiety.

Our response: This is an interesting point that the reviewer mentions as if one of the aminothiols groups demonstrates preferential modification over the other, it can be utilized to append two different labels at two distinct locations in proteins. To test this idea, we produced an MBP variant appended with two 1,2-aminothiol moieties (one at the *N*-terminus and other on the LAP tag) and treated this protein variant with different equivalents of the IBH adduct **1E** under varying reaction conditions. These experiments demonstrated no preferential labelling of either of the two 1,2-aminothiols (details in supporting information Table S16 on page 110 of the revised SI section). However, to accomplish this goal, we developed a step-wise approach (Tandem BHoPAL) to quantitatively generate dually-labelled protein bioconjugates appended with different labels at two distinct specific sites on a single protein molecule. Figure 7b on page 13 of the revised main text depicts this Tandem BHoPAL approach and its use for the dual labelling of proteins (results are discussed on page 14 and all the procedures, supporting figures and tables have been included in the revised version of the supporting file on pages 103–110). We also request the reviewer to read the explanation of our Tandem and Concomitant BHoPAL approaches provided in our response to reviewer #1 above (on page 2 of this document) for more details.

5. One of the major strengths of this method is that it's ability to modify N-Cys proteins even at sub-micromolar concentration (upto 100 nM protein concentration). Did authors try the method to detect naturally occurring N-Cys containing active sites (either known sites or newly discovered sites) in the human proteome?

Our response: We are excited to explore the potential of BHoPAL to profile the *N*-Cys proteome of mammalian cells, are currently exploring this application in our lab.

6. The authors should provide MS/MS data along with the intact mass [C5-1D, C5-1E, C5-1F] & [C7-1D, C7-1E, C7-1F] (any one of them is enough) to further confirm the ligation at the specified position.

Our response: We have performed the MS/MS analysis of one of the MBP-LAP conjugates, **C7-1F**, validating the modification of the protein at its engineered LAP site. This discussion is added on the page 12 of the revised main text, and the corresponding MS/MS data is provided in Fig. S31 of the revised SI section on page 99.

7. The authors should remove the word “diastereoselective” (Fig 1a) and “stereospecific” from the manuscript. Although it is forming exclusively *Z*-isomer after the formation of bis-heterocyclic core, the reaction is not diastereoselective in its actual form (since no new stereocenters are being generated after the reaction). They may call it (*Z*)-selective if they want.

Our response: We thank the reviewer for this suggestion and have replaced these words with “*Z*-selective” throughout the revised main text.

Overall, the manuscript introduces an innovative approach for location-agnostic site-specific protein bioconjugation, effectively overcoming existing limitations. The demonstrated rapid reaction kinetics and broad applicability make it highly valuable for chemical biology and drug development. Addressing the major corrections mentioned above will enhance the clarity and scientific rigor of the manuscript. I recommend acceptance of the paper for publication in *Nature Communications* after the abovementioned revisions.

Our response: We thank the reviewer for this endorsement of our work, and for supporting its publication in *Nature Communications*.

Reviewer #3 (Remarks to the Author):

In this manuscript, a new strategy for the modification of N-terminal Cys residues and 1,2-aminothiol motifs is disclosed using functionalized, isatin-derived Baylis-Hillman adducts. Proof of concept for the reaction of cysteamine with the IBH adducts is explored, confirming reactivity in aqueous media, with reaction rate largely dependent on the pH of the reaction mixture. Chemoselectivity is analyzed through a robustness study in the presence of various amino acids as additives. The reaction is translated to protein examples bearing N-terminal Cys residues exposed via a TEV protease cleavage site. This reaction proceeds efficiently, with stoichiometric or small excesses of IBH reagent, and importantly, is viable at low concentrations (e.g. sub-micromolar range). The rate of reaction is responsive to the nature of the IBH “R” group, with acyl derivatives reacting more rapidly than the corresponding alkyl variants. Labeling of a protein mixture and of live cells expressing N-Cys labelled eGFP is demonstrated in an efficient manner. Finally, a strategy for introducing 1,2-amino thiols into alternative sites within the protein (apart from the N-terminus) is also established which cleverly utilizes the lipoic acid ligase machinery. Optimization of lipoic acid variants featuring a thiazolidine motif is carried out to maximize the yield and rate of incorporation into peptides using the enzyme LplA. Unmasking of the thiazolidine reveals the 1,2-aminothiol motif primed for conjugation. Apart from the IBH chemistry illustrated here, this approach serves to extend the scope and applicability of bioconjugation strategies developed for N-terminal Cys residues.

In my opinion, the work is well-executed, including a detailed supporting information file, and will be of broad interest to the community. I therefore support publication in *Nature Communications*. I would suggest the following revisions prior to publication.

Our response: We thank the reviewer for the endorsement of our work and for supporting its publication in *Nature Communications*.

1. The slow reaction kinetics of existing N-terminal Cys modification strategies is highlighted in the introduction as a major drawback of existing strategies. As a term “slow,” is quite relative. Comparative rate constants for some of the alternative strategies are later discussed (p. 5), but given variability in the measurement and reporting of rate constants from different publications, it would be interesting to know how the IBH reaction fares in a direct competition study, e.g. if cysteamine is treated simultaneously with IBH adduct 1A and an alternative electrophile (e.g. a model thioester – to mimic NCL, with CBT, or cyclopropanone), how much of the desired product is formed? Alternatively, the authors could attempt one of the other approaches under analogous conditions to the IBH chemistry and track reaction progress via HPLC.

Our response: We thank the reviewer for suggesting this experiment and have reported a head-to-head comparison of BHoPAL and different 1,2-aminothiol modification reactions in the revised version of the manuscript. These experiments involved both the one-pot and separate treatments of our IBH adduct **1A**, the cyclopropanone (CPO), and the cyanobenzothiazole (CBT) reagents with cysteamine at neutral pH and at RT. These experiments revealed that the reactivity of **1A** with cysteamine supersedes that of the CPO and CBT reagents. The results of these reactions are reported in Fig. S6 on page 45 of the revised SI and these results are discussed on page 6 of the revised main text.

2. Stability in the presence of external thiols (e.g. glutathione) is evaluated. What happens to the product in the presence of an exogenous 1,2-aminothiol (e.g. cysteamine)? Is there any reversibility observed?

Our response: To evaluate the stability of our bis-heterocyclic conjugate of cysteamine in the presence of *L*-Cys, we incubated our small molecule IBH adduct conjugates **2A** and **2C** with 10-fold molar excess of *L*-Cys. The results demonstrate that **2A** and **2C** retain integrity over 1–2 days' incubation time-period (data is provided in Fig. S8 on page 48 of the revised SI, and these results are discussed on page 6 of the revised main text).

3. In the SI, the stereochemistry of Cys in S2A-Cys should be checked (was this L or D-Cys?); the D-isomer is drawn here, but presumably the more abundant L-amino acid would have been used? The same error is repeated in Scheme S4 (conversion to the activated ester of the thiazolidine) and throughout other thiazolidine examples in the SI.

Our response: We thank the reviewer for pointing out this inadvertent error on our part and have corrected the stereochemistry in all of our *L*-Cys compounds throughout the manuscript.

4. In Table S6 in the SI, uncyclized IBH-POI adduct (B%) is quantified at different time points. However, this is not mentioned in the main text. While this does not necessarily indicate any lack of selectivity for N-terminal as opposed to internal Cys residues (as the N-terminal Gly eGFP, for example, showed no modification), it would still be valuable to comment on this point in the text. Is this initial thiol reaction step reversible, or does the appended amine in the 1,2-aminothiol motif increase the reactivity of the thiol in the first step of the reaction, therefore leading to the formation of the uncyclized IBH-POI conjugate only at the N-terminus?

Our response: The appearance of the uncyclized intermediate (B) was observed only in the case of *N*-alkyl IBH adduct reagents that react slower with 1,2-aminothiols as compared to their *N*-acyl counterparts. As suggested by the reviewer, we have mentioned this observation clearly in the revised main text on page 6 while also citing Table S6. We believe that the cyclization post-Michael addition-elimination is less favourable for *N*-alkyl IBH adducts because of the electron-donating propensity of the alkyl substituent that renders the amidic carbonyl less electrophilic slowing down intramolecular aza-Michael addition-mediated cyclization. We have discussed this point on the first paragraph of page 5 of the main text. We would also like to point out that the issue of reversibility of the thiol addition was also raised by reviewer #1, and we have discussed it in detail above (please see our response to point 3 made by reviewer #1 on page 3 of this document).

REVIEWERS' COMMENTS

Reviewer #1 (Remarks to the Author):

The revised manuscript by Kalia and co-workers provides further characterization to support the 'BHoPAL' bioconjugation method. I appreciate the authors for thoroughly evaluating and implementing my questions/comments/suggestions on a point-by-point basis.

In particular, I am impressed by the addition of Figure 7, which demonstrates dual labelling of proteins using BHoPAL. Furthermore, the competition experiment described in Figure S4 provides more detail into the reaction mechanism. The revised table of contents in the SI assists the reader to navigate this data-rich manuscript more readily.

In summary, my concerns and suggestions have been addressed, and I therefore recommend the manuscript for publication in its current form.

Reviewer #2 (Remarks to the Author)

Based on the reviewers' comments, the Authors have revised the manuscript significantly with additional experiments and data added to the manuscript and the SI. The authors answered all the comments from the reviewers satisfactorily and based on the new data and revised files, I suggest accepting the manuscript without any further delays.

Reviewer #3 (Remarks to the Author):

The revised manuscript satisfactorily addresses all comments from the initial round of review. I appreciate the additional experiments, including the competition studies with alternative thiols and the direct comparison of reaction rates of cysteamine with the IBH adduct versus cysteamine with cyclopropanone (CPO) and cyanobenzothiazole (CBT) reagents. Moreover, the added MS/MS data for the protein conjugates strengthens the argument for 1,2-aminothiol selectivity. Likewise, the authors' inclusion of a strategy for dual-labelling using the devised methodology is impressive; this represents a highly valuable addition to an already comprehensive study. I support publication following a few minor edits.

1. In discussing prior approaches to 1,2-aminothiol labelling, the authors might also consider referencing recent work on cyanopyridine derivatives, e.g. *Angew. Chem. Int. Ed.* 2022, 61, e202114154, and related references.

2. On the bottom of p. 5, the authors discuss the "high nucleophilicity of the sulfhydryl group of cysteamine ($pK_a \sim 8.2$)" in relation to other thiols which have higher pK_a values. I think it is important to note that the higher nucleophilicity of cysteamine is observed at physiological, or near physiological pH, where the cysteamine thiol is deprotonated to a greater extent than the other thiols examined. As nucleophilicity and basicity tend to correlate (with some exceptions), the listing of pK_a values does not

make the rationale for the increased nucleophilicity of cysteamine immediately clear. The pH at which the reactions are performed also plays an important role.

Point-wise response to reviewer's comments (reviewer comments in black and our responses in red).

REVIEWERS' COMMENTS

Reviewer #1 (Remarks to the Author):

The revised manuscript by Kalia and co-workers provides further characterization to support the 'BHoPAL' bioconjugation method. I appreciate the authors for thoroughly evaluating and implementing my questions/comments/suggestions on a point-by-point basis.

In particular, I am impressed by the addition of Figure 7, which demonstrates dual labelling of proteins using BHoPAL. Furthermore, the competition experiment described in Figure S4 provides more detail into the reaction mechanism. The revised table of contents in the SI assists the reader to navigate this data-rich manuscript more readily.

In summary, my concerns and suggestions have been addressed, and I therefore recommend the manuscript for publication in its current form.

Our response: We thank the reviewer for recommending acceptance of our work, and for appreciating our efforts in revising the manuscript.

Reviewer #2 (Remarks to the Author):

Based on the reviewers' comments, the Authors have revised the manuscript significantly with additional experiments and data added to the manuscript and the SI. The authors answered all the comments from the reviewers satisfactorily and based on the new data and revised files, I suggest accepting the manuscript without any further delays.

Our response: We thank the reviewer for recommending acceptance of our work.

Reviewer #3 (Remarks to the Author):

The revised manuscript satisfactorily addresses all comments from the initial round of review. I appreciate the additional experiments, including the competition studies with alternative thiols and the direct comparison of reaction rates of cysteamine with the IBH adduct versus cysteamine with cyclopropanone (CPO) and cyanobenzothiazole (CBT) reagents. Moreover, the added MS/MS data for the protein conjugates strengthens the argument for 1,2-aminothiol selectivity. Likewise, the authors' inclusion of a strategy for dual-labelling using the devised methodology is impressive; this represents a highly valuable addition to an already comprehensive study. I support publication following a few minor edits.

Our response: We thank the reviewer for recommending acceptance of our work, and for appreciating our efforts in revising the manuscript.

1. In discussing prior approaches to 1,2-aminothiol labelling, the authors might also consider referencing recent work on cyanopyridine derivatives, e.g. Angew. Chem. Int. Ed. 2022, 61, e202114154, and related references.

Our response: We thank the reviewer for bringing up this important work. We have cited (as reference no. 60) and discussed this paper in the revised version of the main text (2nd paragraph on page 7 of the main text).

2. On the bottom of p. 5, the authors discuss the “high nucleophilicity of the sulfhydryl group of cysteamine (pKa ~8.2)” in relation to other thiols which have higher pKa values. I think it is important to note that the higher nucleophilicity of cysteamine is observed at physiological, or near physiological pH, where the cysteamine thiol is deprotonated to a greater extent than the other thiols examined. As nucleophilicity and basicity tend to correlate (with some exceptions), the listing of pKa values does not make the rationale for the increased nucleophilicity of cysteamine immediately clear. The pH at which the reactions are performed also plays an important role.

Our response: We thank the reviewer for pointing this out. We have reworded the sentence for better understanding as suggested by the reviewer (4th paragraph of page 4 of the main text).